# SLIGHTLY HARMONIZING CERTIFIED ROBUST RADIUS AND ACCURACY

## ABSTRACT

In the field of certified robustness through randomized smoothing, several works endeavor to improve the certified robust radius through, e.g., examining various smoothing distributions, conducting smooth training with adversarial data, or employing f-divergence based metrics. However, there is a lack of theoretical studies that delve into the relationship between the accuracy performance, the certified robust radius, and the model weights for smoothed classifiers. In the context of this study, we develop a generalization error bound that possesses a certified robust radius for a variant of the smoothed classifier (i.e., the classifier with both smoothed inputs and weights); In other words, the generalization bound holds under any data perturbation within the certified robust radius. As a byproduct, we find that the underpinnings of both the generalization bound and the certified robust radius draw, in part, upon weight spectral norm, which thereby inspires the adoption of spectral regularization in smooth training to boost certified robustness. Utilizing the dimension-independent property of spherical Gaussian inputs in smooth training, we propose a novel and inexpensive spectral regularizer to enhance smoothed classifiers. In addition to the theoretical contribution, an extensive set of empirical results is provided to substantiate the effectiveness of our proposed method.

## 1 INTRODUCTION

Despite their remarkable performance on numerous supervised learning tasks, deep neural networks (DNNs) are often highly vulnerable to adversarial perturbations unnoticeable to the human eye (Szegedy et al., 2013; Goodfellow et al., 2014). This phenomenon has instigated a substantial body of research aimed at enhancing the adversarial robustness of DNNs (Papernot et al., 2016; Tramèr et al., 2017; Xu et al., 2017; Madry et al., 2017; Athalye et al., 2018; Wu et al., 2020). Although notable achievements have been made in the adversarial robustness community, it has been demonstrated that models previously deemed robust have subsequently succumbed to more powerful adversarial attacks (Athalye et al., 2018; Uesato et al., 2018; Croce & Hein, 2020). This has motivated the need for methodologies that provide verifiable guarantees, ensuring that the predictor remains impervious to any attack within a certain perturbation radius. Significant advancements have been achieved in the development of methodologies capable of computing certified robust radius for DNNs (Katz et al., 2017; Wong & Kolter, 2018; Wong et al., 2018; Raghunathan et al., 2018; Huang et al., 2019; Jia et al., 2020), but they demand comprehensive knowledge of the architecture of the predictor and pose challenges in terms of their extensibility to different models.

To address these challenges, recent study has introduced the *randomized smoothing* strategy (Lecuyer et al., 2019; Cohen et al., 2019; Li et al., 2019), an innovative approach aimed at verifying the robustness of smoothed classifiers. Specifically, it employs the smoothing noise to the input data, followed by the determination of the most probable label by the smoothed classifier. Then, the robust radius for the smoothed classifier can be certified. In contrast to other methodologies, randomized smoothing stands out as an efficient and model-agnostic technique, and is highly adaptable to a wide range of machine learning models.

To improve certified robust radius of smoothed classifiers, existing studies have explored numerous smoothing distributions (Lee et al., 2019; Li et al., 2019; Yang et al., 2020; Li et al., 2022), measures of f-divergences (Dvijotham et al., 2020) and adversarial smooth training methods (Salman et al., 2019). However, there is a paucity of research that theoretically studies the aspects of the gener-

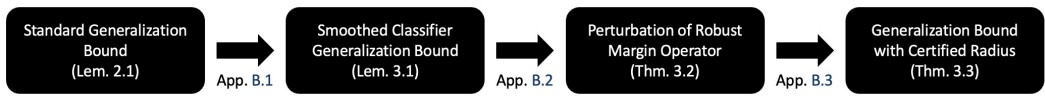

Figure 1: Illustration of the theoretical framework: perturbation bound for smoothed classifiers. Under this framework, a standard generalization bound is extended to a smoothed generalization bound with a certified robust radius.

alization performance, the certified robust radius, and the model weights for smoothed classifiers, which may offer profound insights into the smoothed classifier. To this end, this paper presents a theoretical contribution within the PAC-Bayesian framework, wherein we develop a margin-based generalization bound endowed with a certified robust radius for smoothed classifiers. As shown in Fig. 1, the theoretical development is accomplished through a tripartite process:

1. We extend the existing smoothed classifier, broadening its scope by smoothing both inputs and model weights. Subsequently, we develop a margin-based generalization bound tailored for this new smoothed classifier (Lem. 3.1).

2. Following the margin-based analysis framework, we impose a restriction on the output of the smoothed classifier with respect to its smoothed inputs and weights, proceed to conduct an in-depth generalization analysis leveraging the spectral norm of weights (Thm. 3.2).

3. Building upon the preceding two steps, we derive a certified robust radius for the generalization bound of the smoothed classifier (Thm. 3.3), inspiring us to regularize the spectral norm of weights to boost certified robustness.

In our theoretical setting, we find the bedrock of both the generalization bound and the corresponding certified robust radius derives, in part, from the weight spectral norm. Therefore, we advocate the adoption of an innovative approach to regularize weight spectral norm in smooth training, where smooth training is a popular technique (Cohen et al., 2019; Salman et al., 2019) to enhance smoothed classifiers by applying small smoothing (usually Gaussian distributed) noise to training data. The Gershgorin circle theorem (Gershgorin, 1931) reveals that the weight spectral norm is connected to the scale and cosine similarity of weight vectors. Since the scale of weights has normally been regularized by common techniques like weight decay, this work centers on the regularization of weight cosine similarity. In smooth training with spherical Gaussian inputs, we can handily regularize cosine similarity (and weight spectral norm) by leveraging the $\ell_{1,1}$ entry-wise matrix norm of the output correlation matrix. This scheme offers two key advantages of effectiveness and time-efficiency. Through an extensive set of experiments on a wide range of datasets, we validate the usefulness of our spectral regularization method to boost the certified robustness of smoothed classifiers, with a little extra time consumption. To summarize, the contributions of this work are as follows:

- **Major:** By extending the concepts of margin-based generalization analysis to the smooth setting, we derive a generalization bound that possesses a certified robust radius for smoothed classifiers (Sec. 3).

- **Secondary:** On the strength of theoretical results, we propose an efficient method for spectral regularization in smooth training to enhance certified robustness and provide comprehensive empirical results to demonstrate the effectiveness of our approach (Secs. 4 and 5).

## 2 PRELIMINARIES

**Basic setting.** Consider $\mathbf{x} \in \mathcal{X}$ and $\mathbf{y} \in \mathbb{R}^{d_y}$, where $\mathcal{X} = \{\mathbf{x} \in \mathbb{R}^d | \sum_{i=1}^d x_i^2 \leq B^2\}$ is the input domain and $d_y$ is the number of output targets. Let $\mathbf{y}$ be the one-hot vector of $y$, and $\mathcal{S} = \{(\mathbf{x}_1, y_1), ..., (\mathbf{x}_m, y_m)\}$ represent a training set consisting of $m$ samples drawn independently and identically from an underlying distribution $\mathcal{D}$. Let $\mathcal{F} = \{f_{\mathbf{w}}(\cdot), \mathbf{w} \in \mathcal{W}\}$ represent the learning function class parameterized by $\mathbf{w}$. We define $f_{\mathbf{w}}$ as an $n$-layer neural network with $h$ hidden units per layer with ReLU activation function $\phi(\cdot)$. Each function $f_{\mathbf{w}} : \mathcal{X} \to \mathbb{R}^{d_y}$ in $\mathcal{F}$ maps a data point $\mathbf{x}$ to a $d_y$-dimensional vector. The objective of a supervised learner is to obtain the most effective function in $\mathcal{F}$ that minimizes the expected loss (risk) over $\mathcal{D}$. Let $\mathbf{W}$ and $\mathbf{W}_l$ denote the weight matrices of the model and the $l$-th layer, respectively. Furthermore, let $\mathbf{w}$ and $\mathbf{w}_l$ be the

vectorizations of $\mathbf{W}$ and $\mathbf{W}_l$ (i.e., $\mathbf{w} = \text{vec}(\mathbf{W})$), respectively. We can express each $f_\mathbf{w}(\mathbf{x})$ as $f_\mathbf{w}(\mathbf{x}) = \mathbf{W}_n \phi(\mathbf{W}_{n-1}...\phi(\mathbf{W}_1 \mathbf{x})...)$, and define $f_\mathbf{w}^1(\mathbf{x}) = \mathbf{W}_1 \mathbf{x}$, $f_\mathbf{w}^i(\mathbf{x}) = \mathbf{W}_i \phi(f_\mathbf{w}^{i-1}(\mathbf{x}))$. For convenience, we omit the bias term as it can be incorporated into the weight matrix. The spectral norm of matrix $\mathbf{W}$, denoted as $\|\mathbf{W}\|_2$, represents the largest singular value of $\mathbf{W}$, while the Frobenius norm of $\mathbf{W}$, denoted as $\|\mathbf{W}\|_F$. The $\ell_p$ norm of vector $\mathbf{w}$, denoted as $\|\mathbf{w}\|_p$.

**Margin loss.** The classifier $f_\mathbf{w}$ may exhibit different performance when evaluated on the theoretical data distribution $\mathcal{D}$ and the training dataset $\mathcal{S}$. The generalization error reflects the difference between the average losses observed in the empirical dataset and theoretical data distribution, which are estimated using the training and test datasets, respectively. In prior works such as Neyshabur et al. (2017b); Farnia et al. (2019), the PAC-Bayesian generalization analysis for a DNN is conducted on the margin loss. By considering any positive margin value $\gamma$, the expected margin loss is defined as

$$\mathcal{L}_\gamma(f_\mathbf{w}) := \mathbb{E}_{(\mathbf{x},y)\sim\mathcal{D}} \mathbb{1}\Big[ f_\mathbf{w}(\mathbf{x})[y] \leq \gamma + \max_{j\neq y} f_\mathbf{w}(\mathbf{x})[j] \Big], \tag{1}$$

where $\mathbb{1}[a \leq b] = 1$ if $a \leq b$, else $\mathbb{1}[a \leq b] = 0$. Here setting $\gamma = 0$ corresponds to the normal loss.

**PAC-Bayes.** For randomized predictors, PAC-Bayes (McAllester, 1999; 2003) provides an upper bound on the generalization error with respect to the Kullback-Leibler divergence ($D_{\text{KL}}$) between the posterior distribution $Q$ and the prior distribution $P$ of weights. Neyshabur et al. (2017b); Dziugaite & Roy (2017) consider the posterior distribution $Q$ over predictors of the form $f_{\mathbf{w}+\mathbf{u}^{(w)}}$, where $\mathbf{u}^{(w)}$ is a zero mean Gaussian random variable. For convenience, we denote $\mathbf{w} + \mathbf{u}^{(w)}$ as $\mathbf{w} + \mathbf{u}$. PAC-Bayes can bound the expected loss over posterior distribution $Q$, defined as

$$\mathcal{L}_\gamma(f_{\mathbf{w},\mathbf{u}}) := \mathbb{E}_\mathbf{u}\Big( \mathbb{E}_{(\mathbf{x},y)\sim\mathcal{D}} \mathbb{1}\Big[ f_{\mathbf{w}+\mathbf{u}}(\mathbf{x})[y] \leq \gamma + \max_{j\neq y} f_{\mathbf{w}+\mathbf{u}}(\mathbf{x})[j] \Big] \Big). \tag{2}$$

We then provide a commonly used version of PAC-Bayesian generalization bound in the following.

**Lemma 2.1** *Consider a training dataset $\mathcal{S}$ with $m$ samples drawn from a distribution $\mathcal{D}$ with binary targets. Given a learning algorithm (e.g., a classifier) $f_\mathbf{w}$ with prior and posterior distributions $P$ and $Q$ (i.e., $\mathbf{w} + \mathbf{u}$) on the weights respectively, for any $\delta > 0$, with probability $1 - \delta$ over the draw of training data, we have that*

$$\mathcal{L}_0(f_{\mathbf{w},\mathbf{u}}) \leq \widehat{\mathcal{L}}_0(f_{\mathbf{w},\mathbf{u}}) + 2\sqrt{\frac{2(D_{\text{KL}}(\mathbf{w}+\mathbf{u}\|P) + \log\frac{2m}{\delta})}{m-1}},$$

*where $\mathcal{L}_0(f_{\mathbf{w},\mathbf{u}})$ is the expected loss on $\mathcal{D}$, $\widehat{\mathcal{L}}_0(f_{\mathbf{w},\mathbf{u}})$ is the empirical loss on $\mathcal{S}$, and their difference yields the generalization error.*

**Smoothed classifier.** We consider the smoothed input data as $\mathbf{x} + \mathbf{u}^{(x)}$, where $\mathbf{u}^{(x)}$ is also a zero mean Gaussian random variable. Suppose $\mathbf{u} = \begin{pmatrix} \mathbf{u}^{(w)} \\ \mathbf{u}^{(x)} \end{pmatrix} \sim \mathcal{N}(0, \sigma^2 \mathbf{I})$, we denote $\mathbf{w}+\mathbf{u}^{(w)}, \mathbf{x}+\mathbf{u}^{(x)}$ as $\mathbf{w} + \mathbf{u}$, $\mathbf{x} + \mathbf{u}$ for convenience. Randomized smoothing constructs a smoothed classifier by enhancing a given base classifier with input noise $\mathbf{u}^{(x)}$, where Gaussian distribution is a widely employed smoothing distribution in the literature (Lecuyer et al., 2019; Cohen et al., 2019; Li et al., 2019; Yang et al., 2020). Typically, in these works, the smoothed classifier predicts the class with the highest confidence on the smoothed data, i.e.,

$$\arg\max_{c\in\mathcal{Y}} \mathbb{E}_\mathbf{u} \mathbb{1}\Big[ f_\mathbf{w}(\mathbf{x} + \mathbf{u})[c] > \max_{j\neq c} f_\mathbf{w}(\mathbf{x} + \mathbf{u})[j] \Big]. \tag{3}$$

The technique of smooth training is extensively employed within the framework of randomized smoothing. Specifically, it introduces smoothing noise, often in the form of spherical Gaussian noise, to the training data, which serves to enhance the capability of smoothed classifiers to deal with smoothed inputs.

Different from the previous work, we utilize the smoothed classifier $\mathcal{G}_{\gamma,\mathbf{w},\mathbf{u}}$ with the smoothing distribution on both model weights $\mathbf{w}$ and input data $\mathbf{x}$. When queried at both $\mathbf{w}$ and $\mathbf{x}$, the smoothed classifier $\mathcal{G}_{\gamma,\mathbf{w},\mathbf{u}}$ returns the class that the base classifier $f_\mathbf{w}$ is most likely to select under margin $\gamma$,

$$\mathcal{G}_{\gamma,\mathbf{w},\mathbf{u}}(\mathbf{x}) = \arg\max_{c\in\mathcal{Y}} \begin{cases} \mathbb{E}_\mathbf{u} \mathbb{1}\Big[ f_{\mathbf{w}+\mathbf{u}}(\mathbf{x} + \mathbf{u})[c] > \max_{j\neq c} f_{\mathbf{w}+\mathbf{u}}(\mathbf{x} + \mathbf{u})[j] + \gamma \Big], & \text{if } c = y \\ \mathbb{E}_\mathbf{u} \mathbb{1}\Big[ f_{\mathbf{w}+\mathbf{u}}(\mathbf{x} + \mathbf{u})[c] + \gamma > \max_{j\neq c} f_{\mathbf{w}+\mathbf{u}}(\mathbf{x} + \mathbf{u})[j] \Big]. & \text{if } c \neq y \end{cases} \tag{4}$$

Note that $\gamma$ is a tool that aids in the development of our theory, which allows us to establish an upper bound on the expected loss of $\mathcal{G}_{0,\mathbf{w},\mathbf{u}}$ (i.e., $\gamma = 0$) on $\mathcal{D}$. We then define the expected loss of a smoothed classifier $\mathcal{G}_{\gamma,\mathbf{w},\mathbf{u}}$ as

$$\mathcal{L}_{\gamma}(\mathcal{G}) := \underset{(\mathbf{x},y)\sim\mathcal{D}}{\mathbb{E}}\mathbb{1}\Big[\mathcal{G}_{\gamma,\mathbf{w},\mathbf{u}}(\mathbf{x}) \neq y\Big]. \tag{5}$$

The expected loss under $\ell_2$ norm data perturbation is defined as

$$\mathcal{L}_{\gamma}(\mathcal{G},\epsilon) := \underset{(\mathbf{x},y)\sim\mathcal{D}}{\mathbb{E}}\mathbb{1}\Big[\exists\varepsilon \ \Big| \ \|\varepsilon\|_2^2 \leq \epsilon_{\mathbf{x}}, \ \mathcal{G}_{\gamma,\mathbf{w},\mathbf{u}}(\mathbf{x}+\varepsilon) \neq y\Big], \tag{6}$$

where $\varepsilon \in \mathbb{R}^d$ and $\sqrt{\epsilon_{\mathbf{x}}}$ is the perturbation radius for $\mathbf{x}$. Here we suppose $\epsilon_{\mathbf{x}}$ can be different for different $\mathbf{x}$, which we will discuss later. We consider $\widehat{\mathcal{L}}_{\gamma}(f_{\mathbf{w}})$, $\widehat{\mathcal{L}}_{\gamma}(f_{\mathbf{w},\mathbf{u}})$, $\widehat{\mathcal{L}}_{\gamma}(\mathcal{G})$, $\widehat{\mathcal{L}}_{\gamma}(\mathcal{G},\epsilon)$ to be the empirical estimate of the above expected losses, e.g., $\widehat{\mathcal{L}}_{\gamma}(\mathcal{G})$ is defined as

$$\widehat{\mathcal{L}}_{\gamma}(\mathcal{G}) := \frac{1}{m}\sum_{(\mathbf{x},y)\in\mathcal{S}}\mathbb{1}\Big[\mathcal{G}_{\gamma,\mathbf{w},\mathbf{u}}(\mathbf{x}) \neq y\Big]. \tag{7}$$

The goal of the generalization analysis in this work is to provide theoretical comparison between the true and empirical margin losses for smoothed classifiers. Next, we show how to develop the above PAC-Bayesian generalization analysis for smoothed classifiers, initially bounding $\mathcal{L}_0(\mathcal{G})$ and further extending it to bound $\mathcal{L}_0(\mathcal{G},\epsilon)$ with a certified robust radius $\sqrt{\epsilon_{\mathbf{x}}}$.

## 3 MAIN RESULTS

### 3.1 SKETCH

For clarity, we first outline our principal theoretical result (in Thm. 3.3). Within the PAC-Bayesian framework, we formulate a generalization bound that incorporates a certified robust radius. The primary content is elucidated as follows.

> With probability at least $1 - \delta$, the inequality $\mathcal{L}_0(\mathcal{G},\epsilon) \leq \widehat{\mathcal{L}}_{\gamma}(\mathcal{G}) + \Omega_{\text{ge}}$ holds within the $\ell_2$ norm data perturbation radius $\Omega_{\text{r}}$, both $\Omega_{\text{ge}}$ and $\Omega_{\text{r}}$ are influenced, in part, by $\|\mathbf{W}_i\|_2$.

In other words, this says both the accuracy performance and the certified robust radius, for the smoothed classifier, can be affected by weight spectral norm. Consequently, we have the potential to improve certified robustness through spectral regularization. In the following, we provide the details of our theoretical development, as illustrated in Fig. 1.

### 3.2 ELABORATION

In light of the PAC-Bayesian framework established in Lem. 2.1, which bounds the generalization error over the expected loss of the posterior $Q$, our initial objective is to formulate a generalization bound specifically tailored to the smoothed classifier, as defined in (4) and (5). By leveraging the margin-based generalization analysis approach (Neyshabur et al., 2017b; Bartlett et al., 2017), we have successfully derived a generalization bound for the smoothed classifier through incorporating the empirical margin loss, as elucidated below.

**Lemma 3.1** *Consider Lem. 2.1, let $f_{\mathbf{w}} : \mathcal{X} \to \mathcal{Y}$ denote any predictor with weights $\mathbf{w}$ over training dataset of size $m$, and let $P$ be any prior distribution of weights that is independent of the training data, $\mathbf{w} + \mathbf{u}$ be the posterior of weights. Then, for any $\delta, \gamma > 0$, and any random perturbation $\mathbf{u}$ s.t. $\mathbb{P}_{\mathbf{u}}(\max_{\mathbf{x}}|f_{\mathbf{w}+\mathbf{u}}(\mathbf{x}) - f_{\mathbf{w}}(\mathbf{x})|_{\infty} < \frac{\gamma}{8} \ \cap \ \max_{\mathbf{x}}|f_{\mathbf{w}+\mathbf{u}}(\mathbf{x}+\mathbf{u}) - f_{\mathbf{w}}(\mathbf{x})|_{\infty} < \frac{\gamma}{8}) \geq \frac{1}{2}$, with probability at least $1 - \delta$, we have*

$$\mathcal{L}_0(\mathcal{G}) \leq \widehat{\mathcal{L}}_{\gamma}(\mathcal{G}) + 4\sqrt{\frac{D_{\text{KL}}(\mathbf{w}+\mathbf{u}\|P) + \ln\frac{6m}{\delta}}{m-1}},$$

*where $\mathcal{L}_0(\mathcal{G})$ is the expected loss for the smoothed classifier $\mathcal{G}_{0,\mathbf{w},\mathbf{u}}$ and $\widehat{\mathcal{L}}_{\gamma}(\mathcal{G})$ is the empirical estimate of the margin loss for $\mathcal{G}_{\gamma,\mathbf{w},\mathbf{u}}$.*

*Proof.* See App. B.1. □

Lem. 3.1 follows a similar analysis approach as the margin-based PAC-Bayesian bounds derived for traditional machine learning models (Langford & Shawe-Taylor, 2003; McAllester, 2003) and deep learning models (Neyshabur et al., 2017b; Farnia et al., 2019). However, when considering the lemma in its current form, it constitutes a generalization bound specifically designed for the smoothed classifier. There are two distinctions from previous research: 1. It considers the generalization performance concerning both smoothed weights and data. 2. It employs a $0 - 1$ loss to evaluate the most probable output of the smoothed classifier.

The methodologies employed by Langford & Caruana (2002); Neyshabur et al. (2017a); Dziugaite & Roy (2017) involve utilizing PAC-Bayesian bounds to analyze the generalization behavior of neural networks, wherein the evaluation is conducted either on the KL divergence, the perturbation error $\mathcal{L}_0(f_{\mathbf{w}+\mathbf{u}}) - \mathcal{L}_0(f_{\mathbf{w}})$, or the entire bound numerically. In addition, Neyshabur et al. (2017b); Farnia et al. (2019) employ the PAC-Bayesian framework to derive a margin-based bound that depends on weight norms through restricting $f_{\mathbf{w}+\mathbf{u}}(\mathbf{x}) - f_{\mathbf{w}}(\mathbf{x})$. We will also introduce a margin-based bound that relies on weight norms, but through imposing restrictions on both $f_{\mathbf{w}+\mathbf{u}}(\mathbf{x}) - f_{\mathbf{w}}(\mathbf{x})$ and $f_{\mathbf{w}+\mathbf{u}}(\mathbf{x} + \mathbf{u}) - f_{\mathbf{w}}(\mathbf{x})$ as demonstrated in Lem. 3.1. This is because the smoothed classifier in this work is implemented on both $\mathbf{x} + \mathbf{u}$ and $\mathbf{w} + \mathbf{u}$. As in previous work, we can then choose the largest perturbation $\mathbf{u}$ while adhering to the given constraint to obtain the conditionally tight bound. In the following theorem, we provide the details of the bound, which mainly rests on the spectral norm and the Frobenius norm of weights.

**Theorem 3.2** *Given Lem. 3.1, for any $B, n, h > 0$, let $f_{\mathbf{w}} : \mathcal{X} \to \mathcal{Y}$ be an $n$-layer feedforward network with $h$ units each layer and ReLU activation function. Choose the largest perturbation $\mathbf{u}$ under the restriction, for any $\delta, \gamma > 0$, any $\mathbf{w}$ over training dataset of size $m$, with probability at least $1 - \delta$, we have the the following bound:*

$$\mathcal{L}_0(\mathcal{G}) \le \widehat{\mathcal{L}}_\gamma(\mathcal{G}) + \mathcal{O}\left( \sqrt{\frac{\Phi\left(\prod_i \|\mathbf{W}_i\|_2^2, \sum_i \|\mathbf{W}_i\|_F^2\right) + \ln\frac{nm}{\delta}}{m - 1}} \right),$$

*where*

$$\Phi\left(\prod_i \|\mathbf{W}_i\|_2^2, \sum_i \|\mathbf{W}_i\|_F^2\right) = \frac{\sum_i \left(\|\mathbf{W}_i\|_F^2/\|\mathbf{W}_i\|_2^2\right)}{\Psi\left(\prod_i \|\mathbf{W}_i\|_2^2\right)/\left(\prod_i \|\mathbf{W}\|_2^2\right)^{\frac{1}{n}}},$$

*and*

$$\Psi\left(\prod_i \|\mathbf{W}_i\|_2^2\right) = \left(\left(\gamma \Big/ \left(2^8 n(h\ln(8nh))^{\frac{1}{2}} \tau^{\frac{1}{2}} \prod_i \|\mathbf{W}_i\|_2^{\frac{n-1}{n}}\right) + \frac{B^2}{4\tau}\right)^{\frac{1}{2}} - \frac{B}{2\tau^{\frac{1}{2}}}\right)^2.$$

*Here $\tau$ is the solution of $F_{\chi_d^2}(\tau) = \frac{\sqrt{2}}{2}$, where $F_{\chi_d^2}(\cdot)$ is the cumulative distribution function (CDF) for the chi-square distribution $\chi_d^2$ with $d$ degrees of freedom.*

*Proof.* See App. B.2. □

The above theorem derives the generalization guarantee by restricting the variation in the output of the network and thus bounding the sharpness of the model. This approach is similar to Neyshabur et al. (2017b); Bartlett et al. (2017), yet with some notable differences: the previous work depicts the change of the output with respect to smoothed weights, whereas we consider the variation of the output in connection with both smoothed weights and smoothed inputs for the smoothed classifier.

Next, we extend Thm. 3.2 to a certified robustness setting. Although randomized smoothing has been extensively explored in previous research to provide robustness guarantee for a specific input, limited attention has been given to the relationship between the certified robust radius, the generalization performance, and the model weights. As a first attempt, we endeavor to develop the generalization bound under a certified robust radius, i.e., the generalization bound holds for any data perturbation within the radius.

**Theorem 3.3** *Given Thm. 3.2, for any* $\mathbf{x} \in \mathcal{X}$, *suppose there exist* $p_{\mathbf{w},\mathbf{u}}^A(\mathbf{x})$, $p_{\mathbf{w},\mathbf{u}}^B(\mathbf{x})$ *such that*

$$\mathbb{E}_{\mathbf{u}} \mathbb{1}\left[ \arg\max_c f_{\mathbf{w}+\mathbf{u}}(\mathbf{x}+\mathbf{u})[c] = \mathcal{G}_{0,\mathbf{w},\mathbf{u}}(\mathbf{x}) \right] \geq p_{\mathbf{w},\mathbf{u}}^A(\mathbf{x}) \geq p_{\mathbf{w},\mathbf{u}}^B(\mathbf{x})$$

$$\geq \max_{j \neq \mathcal{G}_{0,\mathbf{w},\mathbf{u}}(\mathbf{x})} \mathbb{E}_{\mathbf{u}} \mathbb{1}\left[ \arg\max_c f_{\mathbf{w}+\mathbf{u}}(\mathbf{x}+\mathbf{u})[c] = j \right].$$

*Then, for any* $\delta, \gamma > 0$, *with probability at least* $1-\delta$ *we have*

$$\mathcal{L}_0(\mathcal{G}, \epsilon) \leq \widehat{\mathcal{L}}_\gamma(\mathcal{G}) + \mathcal{O}\left( \sqrt{ \frac{\Phi\left( \prod_i \|\mathbf{W}_i\|_2^2, \sum_i \|\mathbf{W}_i\|_F^2 \right) + \ln \frac{nm}{\delta}}{m-1} } \right)$$

*within* $\ell_2$ *norm data perturbation radius* $\sqrt{\epsilon_{\mathbf{x}}}$, *where*

$$\epsilon_{\mathbf{x}} = \underbrace{-\ln\left( 1 - \left( \sqrt{p_{\mathbf{w},\mathbf{u}}^A(\mathbf{x})} - \sqrt{p_{\mathbf{w},\mathbf{u}}^B(\mathbf{x})} \right)^2 \right)}_{\textbf{\textit{Model and Data Joint Dependence}}} \cdot \underbrace{2 \Psi\left( \prod_i \|\mathbf{W}_i\|_2^2 \right)}_{\textbf{\textit{Model Dependence}}}. \tag{8}$$

*Proof.* See App. B.3. □

In Thm. 3.3, we utilize a commonly employed setup of randomized smoothing that there exists a lower bound $p_{\mathbf{w},\mathbf{u}}^A(\mathbf{x})$ of the largest probability of the output and an upper bound $p_{\mathbf{w},\mathbf{u}}^B(\mathbf{x})$ of the secondary probability of the output. The gap between $\sqrt{p_{\mathbf{w},\mathbf{u}}^A(\mathbf{x})}$ and $\sqrt{p_{\mathbf{w},\mathbf{u}}^A(\mathbf{x})}$ determines the first component of the certified robust radius in (8), which is jointly influenced by both the model and the input data. In addition, the second component of the certified robust radius depends solely on the model, primarily on the spectral norm of weights.

The above theorem illustrates that, within our theoretical framework for the smoothed classifier, both the generalization bound and its corresponding certified robust radius are partially influenced by the spectral norm of weights. Assume other factors remain constant, reducing the spectral norm of weights can, generally and simultaneously, narrow the generalization bound and enhance the certified robust radius. Thus, Thm. 3.3 leads to our conclusion that the spectral regularization of weights may enhance the smoothed classifier in the aspect of certified robust radius and generalization performance. This theoretical analysis will guide our design of regularization in the next section for smooth training.

## 4 SPECTRAL REGULARIZATION IN SMOOTH TRAINING

Thm. 3.3 reveals the existence of two components in (8) that influence the certified robust radius. The first component pertains to the gap between $\sqrt{p_{\mathbf{w},\mathbf{u}}^A(\mathbf{x})}$ and $\sqrt{p_{\mathbf{w},\mathbf{u}}^B(\mathbf{x})}$, which depends on both the model and the data. The second component, denoted as $\Psi(\prod_i \|\mathbf{W}_i\|_2^2)$, is solely reliant on the model, mainly on the spectral norm of model weights. In previous work (Cohen et al., 2019; Salman et al., 2019), smooth training is designed to augment the first component of the gap through adding Gaussian noise to the training data. From a different point of view, Thm. 3.3 in this work reaches a theoretical perspective that the weight spectral norm in the second component may also be a key factor to the generalization and certified robust radius — our empirical results in Sec. 5 will provide further evidence. Thus in this section, we will explore how the spectral norm can be utilized to help smooth training to learn a better smoothed classifier.

In previous literature, spectral regularization has been used to improve generalization (Yoshida & Miyato, 2017) and adversarial robustness (Farnia et al., 2019). However, considering the inherent difficulty in computing the spectral norm, estimating it for each layer of weights is still a costly endeavor. In this work, thanks to the spherical Gaussian inputs in smooth training, we can harness the correlation matrix to craft an economical approach for spectral regularization as follows.

Given $\mathbf{W} = \mathbf{W}_n \mathbf{W}_{n-1} \cdots \mathbf{W}_1$, according to the Gershgorin circle theorem, we have

$$\|\mathbf{W}\|_2^2 \leq \|\mathbf{W}\mathbf{W}^\top\|_\infty = \max_i \sum_j \left[ \underbrace{\|\mathbf{w}^{(i)}\|_2 \|\mathbf{w}^{(j)}\|_2}_{\text{Scale}} \underbrace{|\cos(\mathbf{w}^{(i)}, \mathbf{w}^{(j)})|}_{\text{Correlation}} \right], \tag{9}$$

where $\mathbf{w}^{(i)}$ is the $i$-th row vector of $\mathbf{W}$.

The scale term presented in (9) has been extensively studied in previous research, particularly in seminal studies on weight decay (LeCun et al., 2015; Loshchilov & Hutter, 2017) and batch normalization (Ioffe & Szegedy, 2015), among other notable works. In this work, given the weight scale has normally been regularized (e.g., via weight decay), we propose to leverage the correlation term in (9) to impose regularization on $\|\mathbf{W}\|_2$ and, furthermore, $\prod_i \|\mathbf{W}_i\|_2$. Although minimizing the correlation term, also known as weight orthogonality (Saxe et al., 2013; Mishkin & Matas, 2015; Bansal et al., 2018), has been the subject of prior research, our algorithm is different from previous work in two aspects: Firstly, the regularization is directly applied to the whole weight matrix $\mathbf{W}$, which is an ensemble of all layer weight matrices; Secondly, our algorithm is specially crafted to leverage the spherical Gaussian inputs for spectral regularization, leading to a little additional time consumption (see Tab. 1).

In addition, we would like to point out that optimizing the correlation term for spectral regularization is a worthwhile endeavor. Suppose the weight scale in (9) remains invariant (e.g., after regularization by weight decay), both $|\cos(\mathbf{w}^{(i)}, \mathbf{w}^{(j)})|$ and $\|\mathbf{W}\|_2^2$ can then achieve the optimum simultaneously. That is, given a weight matrix $\mathbf{W}$ (with fixed $\|\mathbf{w}^{(i)}\|_2$) s.t. $\forall i \neq j$, $\cos(\mathbf{w}^{(i)}, \mathbf{w}^{(j)}) = 0$, we have $\|\mathbf{W}\|_2^2 = \|\mathbf{W}\mathbf{W}^\top\|_\infty$ and $\|\mathbf{W}\|_2^2$ attains its minimum for the given scale. Moreover, the empirical results in Sec. 5.1 demonstrate that regularizing the correlation term in (9) yields a notable reduction in both $\|\mathbf{W}\|_2$ and $\prod_i \|\mathbf{W}_i\|_2$. In the following, we will provide the details of our novel and cheap method for spectral regularization in smooth training.

Let $\widetilde{f}_\mathbf{w}$ be the linear (remove non-linear components) classifier parameterized by $\mathbf{W}$ and $\widetilde{f}_\mathbf{w}(\mathbf{x})[i]$ be the $i$-th output of $\widetilde{f}_\mathbf{w}(\mathbf{x})$. Given $\boldsymbol{\eta} \in \mathbb{R}^d$ is a spherical Gaussian, the cosine similarity between $\mathbf{w}^{(i)}$ and $\mathbf{w}^{(j)}$ is equal to the statistical correlation (Pearson correlation coefficient) between $\widetilde{f}_\mathbf{w}(\mathbf{x}+\boldsymbol{\eta})[i]$ and $\widetilde{f}_\mathbf{w}(\mathbf{x}+\boldsymbol{\eta})[j]$, i.e.,

$$\cos(\mathbf{w}^{(i)}, \mathbf{w}^{(j)}) = \rho(\widetilde{f}_\mathbf{w}(\mathbf{x}+\boldsymbol{\eta})[i], \widetilde{f}_\mathbf{w}(\mathbf{x}+\boldsymbol{\eta})[j])$$
$$= \frac{\mathbb{E}_{\boldsymbol{\eta}}[(\widetilde{f}_\mathbf{w}(\mathbf{x}+\boldsymbol{\eta})[i] - \mathbb{E}_{\boldsymbol{\eta}}(\widetilde{f}_\mathbf{w}(\mathbf{x}+\boldsymbol{\eta})[i]))(\widetilde{f}_\mathbf{w}(\mathbf{x}+\boldsymbol{\eta})[j] - \mathbb{E}_{\boldsymbol{\eta}}(\widetilde{f}_\mathbf{w}(\mathbf{x}+\boldsymbol{\eta})[j]))]}{\sqrt{\mathbb{E}_{\boldsymbol{\eta}}[(\widetilde{f}_\mathbf{w}(\mathbf{x}+\boldsymbol{\eta})[i] - \mathbb{E}_{\boldsymbol{\eta}}(\widetilde{f}_\mathbf{w}(\mathbf{x}+\boldsymbol{\eta})[i]))^2]\mathbb{E}_{\boldsymbol{\eta}}[(\widetilde{f}_\mathbf{w}(\mathbf{x}+\boldsymbol{\eta})[j] - \mathbb{E}_{\boldsymbol{\eta}}(\widetilde{f}_\mathbf{w}(\mathbf{x}+\boldsymbol{\eta})[j]))^2]}}. \quad (10)$$

Note that here $\mathbf{x}$ is a constant vector and $\boldsymbol{\eta}$ is a random vector. Let $\mathcal{R}(\widetilde{f}_\mathbf{w})$ be the correlation matrix of random vector $\widetilde{f}_\mathbf{w}(\mathbf{x}+\boldsymbol{\eta})$, i.e., $\rho(\widetilde{f}_\mathbf{w}(\mathbf{x}+\boldsymbol{\eta})[i], \widetilde{f}_\mathbf{w}(\mathbf{x}+\boldsymbol{\eta})[j])$ is the element of $i$-th row and $j$-th column of $\mathcal{R}(\widetilde{f}_\mathbf{w})$, we thus have

$$\sum_i \sum_j |\cos(\mathbf{w}^{(i)}, \mathbf{w}^{(j)})| = \|\mathcal{R}(\widetilde{f}_\mathbf{w})\|_{1,1}, \quad (11)$$

where $\|\cdot\|_{1,1}$ is the $\ell_{1,1}$ entry-wise matrix norm, represents the sum of absolute elements in the matrix. Therefore, to minimize the spectral norm of $\mathbf{W}$ through correlation term in (9), we can incorporate the regularization term $\|\mathcal{R}(\widetilde{f}_\mathbf{w})\|_{1,1}$ into the objective function of smooth training, i.e.,

$$\mathcal{L}(f(\mathbf{x}+\boldsymbol{\eta}), y) + \alpha \cdot \|\mathcal{R}(\widetilde{f}_\mathbf{w})\|_{1,1}, \quad (12)$$

where $\alpha \in [0, +\infty)$ is a hyper-parameter to balance the relative contributions of smooth training loss $\mathcal{L}(f(\mathbf{x}+\boldsymbol{\eta}), y)$ and regularization term $\|\mathcal{R}(\widetilde{f}_\mathbf{w})\|_{1,1}$. For time-efficiency, we just compute $\|\mathcal{R}(\widetilde{f}_\mathbf{w})\|_{1,1}$ once per epoch. In the next section, we will provide extensive empirical results to demonstrate that our method can successfully regularize weight spectral norm and enhance smoothed classifiers.

## 5 EMPIRICAL RESULTS

Within this section, we present a comprehensive experiment with the purpose of substantiating the derived bound in Sec. 3 as well as validating the effectiveness of our spectral regularization method as delineated in Sec. 4. Our results illustrate that the spectral regularization method enhances certified robustness for smoothed classifiers across diverse datasets. Specifically, in Sec. 5.1, we demonstrate that our regularizer can effectively reduce the weight spectral norm with a little extra time consumption. In Sec. 5.2, we show the reduced weight spectral norm can enhance certified robustness within our evaluation framework.

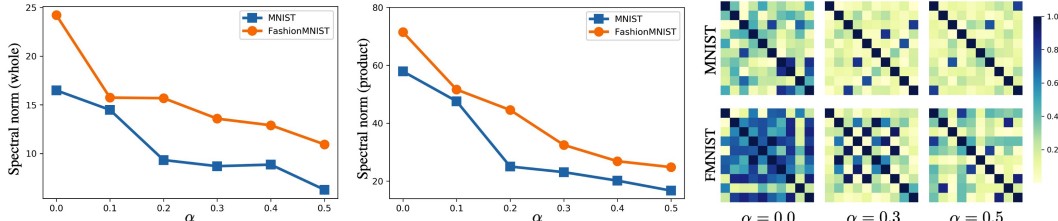

Figure 2: We train MLPs on MNIST and FashionMNIST with $\alpha \in \{0.0, 0.1, 0.2, 0.3, 0.4, 0.5\}$, respectively. **Left:** the spectral norm of the whole weight matrix, i.e., $\|\mathbf{W}\|_2$ where $\mathbf{W} = \mathbf{W}_n \mathbf{W}_{n-1} \cdots \mathbf{W}_1$, with respect to $\alpha$. **Middle:** the product of the spectral norms of the weight matrix, i.e., $\prod_i \|\mathbf{W}_i\|_2$, with respect to $\alpha$. **Right:** the cosine similarity matrix of row vectors of the whole weight matrix $\mathbf{W}$.

Table 1: Running time (seconds/epoch) for MLP on MNIST, ResNet110 on CIFAR-10 and ResNet50 on ImageNet.

| MLP (MNIST) | | | ResNet110 (CIFAR-10) | | | ResNet50 (ImageNet) | | |
|---|---|---|---|---|---|---|---|---|
| Normal | Spectral | Gap | Normal | Spectral | Gap | Normal | Spectral | Gap |
| 17.0 | 17.1 | +0.5% | 32.6 | 33.7 | +3.3% | 8970 | 9160 | +2.1% |

## 5.1 Spectral Regularization

Since the weight spectral norm of small multilayer perceptron (MLP) can be easily computed, we implement MLPs with different values of hyper-parameter $\alpha$ on MNIST and FashionMNIST to show the efficacy of our approach in reducing weight spectral norm. As shown in Fig. 2, our method can effectively reduce the whole weight matrix spectral norm ($\|\mathbf{W}\|_2$) and the product of weight spectral norms ($\prod_i \|\mathbf{W}_i\|_2$). Both of them significantly decrease as $\alpha$ increases. However, large $\alpha$ can also damage the training accuracy of the classifier, thus we set $\alpha = 0.1$ as the default value for the following experiments in Sec. 5.2. Moreover, our method incurs only a minor overhead in training time due to the ease of computing the correlation matrix in a linear network, as demonstrated in Tab. 1.

## 5.2 Enhance Certified Robustness

In adversarially robust classification, one important metric is the certified test accuracy at radius $r$. This metric is defined as the fraction of the test set that $\mathcal{G}_{0,\mathbf{w},\mathbf{u}}$ classifies correctly with a prediction that is certifiably robust within an $\ell_2$ norm ball of radius $r$. Following Cohen et al. (2019), we also report the approximate certified test accuracy with respect to the radius. When running the evaluation algorithm from Cohen et al. (2019), we use 100 Monte Carlo samples for selection, 100000 samples for estimation on MNIST (LeCun & Cortes, 2010) and FashionMNIST, 10000 samples for estimation on CIFAR-10 (Krizhevsky et al., 2009) and ImageNet (Deng et al., 2009). Our evaluation method differs from previous work in that we use a certified framework that is tailored to our smoothed classifiers for both smoothed inputs and weights (details are given in App. C). Specifically, the certified robust radius in our evaluation is defined as

$$R = \sqrt{-2\sigma^2 \ln\left(1 - \left(\sqrt{p^A_{\mathbf{w},\mathbf{u}}(\mathbf{x})} - \sqrt{p^B_{\mathbf{w},\mathbf{u}}(\mathbf{x})}\right)^2\right)}, \tag{13}$$

where $p^A_{\mathbf{w},\mathbf{u}}(\mathbf{x})$, $p^B_{\mathbf{w},\mathbf{u}}(\mathbf{x})$ share the same definitions as outlined in Thm. 3.3, and $\sigma^2$ represents the variance of the smoothing noise. Details are given in Prop. 5.2 in App. B.4. In contrast, Cohen et al. (2019); Salman et al. (2019) employ the evaluation that is only designed for smoothed inputs.

Fig. 3 plots the upper envelope of the certified accuracies for MLPs trained on MNIST and FashionMNIST, as well as ResNet110 (He et al., 2016) on CIFAR-10 and ResNet50 on ImageNet. For the normal trained models, we set $\sigma = 0.12$, while for the spectral regularized models, we use $\sigma = 0.13$. As shown in Fig. 3, the application of spectral regularization consistently leads to superior certified robustness performance across all four datasets. The spectral regularized models exhibit the ability to maintain high levels of accuracy while slightly increasing the certified robust radius by elevating $\sigma$.

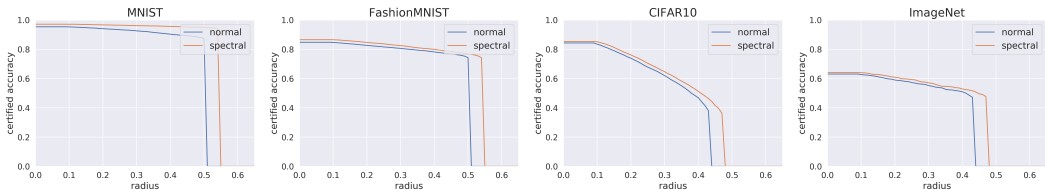

Figure 3: Experiments with randomized smoothing on both weights and data, for MLP on MNIST, MLP on FashionMNIST, ResNet110 on CIFAR-10, and ResNet50 on ImageNet, respectively. We certify the full MNIST, FashionMNIST, CIFAR-10 test sets and a subsample of 1000 examples from the ImageNet test set. We choose $\sigma = 0.12$ for normal trained models and $\sigma = 0.13$ for spectral regularized models.

## 6 RELATED WORK

**PAC-Bayes** is a general framework that enables efficient study of generalization for various machine learning models. Numerous canonical works have been developed in the past decades since the inception of PAC-Bayes by McAllester (1999). E.g., Seeger (2002) studies PAC-Bayes for Gaussian process classification; Langford & Caruana (2002) uses PAC-Bayes to bound the true error rate of a continuous valued classifier; Langford & Shawe-Taylor (2003) develops a margin-based PAC-Bayesian bound which is more data-dependent; Germain et al. (2009) proposes different learning algorithms to find linear classifiers that minimize PAC-Bayesian bounds; Parrado-Hernández et al. (2012) explores the capabilities of PAC-Bayes to provide tight predictions for the generalization of SVM; Alquier et al. (2016) shows that when the risk function is convex, a variational approximation of PAC-Bayesian posterior can be obtained in polynomial time; Thiemann et al. (2017) proposes a PAC-Bayesian bound and a way to construct a hypothesis space, so that the bound is convex in the posterior distribution; Dziugaite & Roy (2018) shows a differentially private data-dependent prior can yield a valid PAC-Bayesian bound; Letarte et al. (2019) studies PAC-Bayes for multilayer neural networks with binary activation; Rivasplata et al. (2020) provides a basic PAC-Bayesian inequality for stochastic kernels; Jin et al. (2020) studies PAC-Bayes with non-spherical Gaussian posterior distribution for DNNs; Pérez-Ortiz et al. (2021) conducts an empirical study on the use of training objectives derived from PAC-Bayesian bounds to train probabilistic neural networks; Dziugaite et al. (2021) shows that in some cases, a stronger PAC-Bayesian bound can be obtained by using a data-dependent oracle prior; Lotfi et al. (2022) develops a compression approach to provide tight PAC-Bayesian generalization bounds on a variety of tasks, including transfer learning; Haddouche & Guedj (2022) proves new PAC-Bayesian bounds in the online learning framework; Amit et al. (2022) presents a PAC-Bayesian generalization bound which enables the replacement of the KL divergence with a variety of Integral Probability Metrics; Wu & Seldin (2022) derives a PAC-Bayes-split-kl inequality to bound the expected loss; Biggs & Guedj (2023) introduces a modified version of the excess risk to obtain empirically tighter, faster-rate PAC-Bayesian generalization bounds. Livni & Moran (2020) presents a limitation for the PAC-Bayes and demonstrates an easy learning task that is not amenable to a PAC-Bayesian analysis. More related work on **randomized smoothing** is provided in App. A.

In contrast to previous research, this paper investigates the theoretical properties of smoothed classifiers within the PAC-Bayesian framework. As shown in Fig. 1, the most notable difference is that this work introduces a margin-based generalization bound tailored for the smoothed classifier with a certified robust radius.

## 7 CONCLUSION

In this paper, we explored the theoretical connection between the generalization performance, the certified robust radius, and the model weights for smoothed classifiers, leading to the development of a margin-based generalization bound within our smooth framework. As a result of these theoretical findings, we were inspired to formulate a highly effective and efficient spectral regularizer tailored for smooth training. Subsequently, we conducted extensive experiments to empirically demonstrate the effectiveness of our spectral regularizer in improving certified robustness.

**Reproducibility Statement:** We provide the proofs for Sec. 3 in App. B, the code for the experiments in the supplementary file.

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

The appendices can be summarized as follows:

- App. A: In this section, we provide more related work about randomized smoothing.
- App. B: We provide the detailed proofs for Lem. 3.1, Thm. 3.2, Thm. 3.3 and Prop. 5.2.
- App. C: We present the details of the evaluation in Sec. 5.

## A MORE RELATED WORK

Following the groundbreaking research of **randomized smoothing** by Lecuyer et al. (2019); Cohen et al. (2019); Li et al. (2019), several notable contributions have emerged. Kumar et al. (2020b) shows that extending the smoothing technique to defend against other attack models can be challenging. Lee et al. (2019); Schuchardt et al. (2020); Wang et al. (2021) introduce extensions of randomized smoothing that address discrete perturbations like $\ell_0$-perturbations, whereas Bojchevski et al. (2020); Gao et al. (2020); Levine & Feizi (2020b); Liu et al. (2021) delve into extensions related to graphs, patches, and point cloud manipulations. Yang et al. (2020); Zhang et al. (2020) derive methods to determine certificates with $\ell_1, \ell_2$, and $\ell_\infty$ norms. Xie et al. (2021) trains certifiably robust federated learning models against backdoor attacks via randomized smoothing. Mu et al. (2023) leverages randomized smoothing for multi-agent reinforcement learning system to determine actions with guaranteed certified bounds. Dvijotham et al. (2020) offers theoretical derivations applicable to both continuous and discrete smoothing measures, whereas Mohapatra et al. (2020) enhances certificates by incorporating gradient information and Horváth et al. (2021) uses ensembles to improve certificates. Rosenfeld et al. (2020); Levine & Feizi (2020a); Jia et al. (2021); Weber et al. (2023) use randomized smoothing to defend against data poisoning attacks. In addition to norm-balls certificates, Fischer et al. (2020); Li et al. (2021) demonstrate how randomized smoothing can be used to certify geometric operations like rotation and translation. In their respective works, Chiang et al. (2020); Fischer et al. (2021) show how certificates can be extended from classification to regression (and object detection) and segmentation. In the context of classification, Jia et al. (2020) broadens the scope of certificates by encompassing not only the top-1 class but also the top-k classes, whereas Kumar et al. (2020a) focuses on certifying the confidence of the classifier, extending beyond solely the top-class prediction. Hong et al. (2022) proposes a universally approximated certified robustness framework to provide robustness certification for any input data on any classifier; Hao et al. (2022) proposes generalized randomized smoothing to certify robustness against general semantic transformations. Schuchardt & Günnemann (2022) proposes a gray-box approach to enhance randomized smoothing technique with white-box knowledge about invariances. Mehra et al. (2021) presents a novel bilevel optimization-based poisoning attack that damages the robustness of certified robust models. Alfarra et al. (2022b) shows that the variance of the Gaussian distribution in randomized smoothing can be optimized at each input so as to maximize the certification radius. Mohapatra et al. (2021) shows that the decision boundaries of smoothed classifiers will shrink, leading to disparity in class-wise accuracy. Alfarra et al. (2022a) reformulates the randomized smoothing framework which can scale to large networks on large input datasets. Chen et al. (2022) proposes Input-Specific Sampling acceleration to achieve the cost-effectiveness for randomized smoothing robustness certification.

In contrast to previous work on randomized smoothing, this paper explores the theoretical properties of smoothed classifiers under PAC-Bayesian framework. Specifically, it offers a theoretical explanation of how a smoothed classifier can achieve superior generalization performance and a larger certified robust radius.

# B   PROOFS

## B.1   PROOF FOR LEM. 3.1

First, we restate Lem. 3.1 here.

**Lemma 3.1** *Consider Lem. 2.1, let $f_{\mathbf{w}} : \mathcal{X} \to \mathcal{Y}$ denote any predictor with weights $\mathbf{w}$ over training dataset of size $m$, and let $P$ be any prior distribution of weights that is independent of the training data, $\mathbf{w} + \mathbf{u}$ be the posterior of weights. Then, for any $\delta, \gamma > 0$, and any random perturbation $\mathbf{u}$ s.t. $\mathbb{P}_{\mathbf{u}}(\max_{\mathbf{x}} |f_{\mathbf{w}+\mathbf{u}}(\mathbf{x}) - f_{\mathbf{w}}(\mathbf{x})|_{\infty} < \frac{\gamma}{8} \ \cap \ \max_{\mathbf{x}} |f_{\mathbf{w}+\mathbf{u}}(\mathbf{x}+\mathbf{u}) - f_{\mathbf{w}}(\mathbf{x})|_{\infty} < \frac{\gamma}{8}) \geq \frac{1}{2}$, with probability at least $1 - \delta$, we have*

$$\mathcal{L}_0(\mathcal{G}) \leq \widehat{\mathcal{L}}_{\gamma}(\mathcal{G}) + 4\sqrt{\frac{D_{\mathrm{KL}}(\mathbf{w}+\mathbf{u}\|P) + \ln\frac{6m}{\delta}}{m-1}},$$

*where $\mathcal{L}_0(\mathcal{G})$ is the expected loss for the smoothed classifier $\mathcal{G}_{0,\mathbf{w},\mathbf{u}}$ and $\widehat{\mathcal{L}}_{\gamma}(\mathcal{G})$ is the empirical estimate of the margin loss for $\mathcal{G}_{\gamma,\mathbf{w},\mathbf{u}}$.*

In the following, we provide the details of the proof.

**Proof for Lem. 3.1.** *Let $\mathcal{S}_{\mathbf{u}}$ be the set of perturbations with the following property:*

$$\mathcal{S}_{\mathbf{u}} \subseteq \left\{ \mathbf{u} \,\Big|\, \max_{\mathbf{x}\in\mathcal{X}} |f_{\mathbf{w}+\mathbf{u}}(\mathbf{x}+\mathbf{u}) - f_{\mathbf{w}}(\mathbf{x})|_{\infty} < \frac{\gamma}{8} \ \cap \ \max_{\mathbf{x}\in\mathcal{X}} |f_{\mathbf{w}+\mathbf{u}}(\mathbf{x}) - f_{\mathbf{w}}(\mathbf{x})|_{\infty} < \frac{\gamma}{8} \right\}. \quad (14)$$

*Let $q$ be the probability density function over $\mathbf{u}$. We construct a new distribution $\tilde{Q}$ over $\tilde{\mathbf{u}}$ that is restricted to $\mathcal{S}_{\mathbf{u}}$ with the probability density function:*

$$\tilde{q}(\tilde{\mathbf{u}}) = \frac{1}{z} \begin{cases} q(\tilde{\mathbf{u}}) & \tilde{\mathbf{u}} \in \mathcal{S}_{\mathbf{u}}, \\ 0 & otherwise, \end{cases} \quad (15)$$

*where $z$ is a normalizing constant and by the lemma assumption $z = \mathbb{P}(\mathbf{u} \in \mathcal{S}_{\mathbf{u}}) \geq \frac{1}{2}$. By the definition of $\tilde{Q}$, we have:*

$$\max_{\mathbf{x}\in\mathcal{X}} |f_{\mathbf{w}+\tilde{\mathbf{u}}}(\mathbf{x}+\tilde{\mathbf{u}}) - f_{\mathbf{w}}(\mathbf{x})|_{\infty} < \frac{\gamma}{8} \ and \ \max_{\mathbf{x}\in\mathcal{X}} |f_{\mathbf{w}+\tilde{\mathbf{u}}}(\mathbf{x}) - f_{\mathbf{w}}(\mathbf{x})|_{\infty} < \frac{\gamma}{8}. \quad (16)$$

*Therefore, with probability at least $1 - \delta$ over training dataset $\mathcal{S}$, we have:*

$$
\begin{aligned}
\mathcal{L}_0(\mathcal{G}) &\leq \mathcal{L}_{\frac{\gamma}{2}}(f_{\mathbf{w},\tilde{\mathbf{u}}}) && \triangleright because\ of\ Pf.\ B.1 \\
&\leq \widehat{\mathcal{L}}_{\frac{\gamma}{2}}(f_{\mathbf{w},\tilde{\mathbf{u}}}) + 2\sqrt{\frac{2(D_{\mathrm{KL}}(\mathbf{w}+\tilde{\mathbf{u}}\|P) + \log\frac{2m}{\delta})}{m-1}} && \triangleright because\ of\ Lem.\ 2.1 \\
&\leq \widehat{\mathcal{L}}_{\gamma}(\mathcal{G}) + 2\sqrt{\frac{2(D_{\mathrm{KL}}(\mathbf{w}+\tilde{\mathbf{u}}\|P) + \log\frac{2m}{\delta})}{m-1}} && \triangleright because\ of\ Pf.\ B.2 \\
&\leq \widehat{\mathcal{L}}_{\gamma}(\mathcal{G}) + 4\sqrt{\frac{D_{\mathrm{KL}}(\mathbf{w}+\mathbf{u}\|P) + \log\frac{6m}{\delta}}{m-1}} && \triangleright because\ of\ Pf.\ B.3
\end{aligned}
$$

*Hence, proved.* $\qquad\square$

**Proof B.1** *Given (14) and (15), for all $\tilde{\mathbf{u}} \in \tilde{Q}$, we have*

$$
\begin{aligned}
\max_{\mathbf{x}\in\mathcal{X}} |f_{\mathbf{w}+\tilde{\mathbf{u}}}(\mathbf{x}+\tilde{\mathbf{u}}) - f_{\mathbf{w}}(\mathbf{x})|_{\infty} &< \frac{\gamma}{8}, \\
\max_{\mathbf{x}\in\mathcal{X}} |f_{\mathbf{w}+\tilde{\mathbf{u}}}(\mathbf{x}) - f_{\mathbf{w}}(\mathbf{x})|_{\infty} &< \frac{\gamma}{8}.
\end{aligned} \quad (17)
$$

*For all $\mathbf{x} \in \mathcal{X}$ s.t. $\mathcal{G}_{0,\mathbf{w},\mathbf{u}}(\mathbf{x}) \neq y$, as $z = \mathbb{P}(\mathbf{u} \in \mathcal{S}_{\mathbf{u}}) \geq \frac{1}{2}$, there exists $\check{\mathbf{u}} \in \tilde{Q}$ s.t.*

$$f_{\mathbf{w}+\check{\mathbf{u}}}(\mathbf{x}+\check{\mathbf{u}})[\mathcal{G}_{0,\mathbf{w},\mathbf{u}}(\mathbf{x})] > f_{\mathbf{w}+\check{\mathbf{u}}}(\mathbf{x}+\check{\mathbf{u}})[y]. \quad (18)$$

Then for all $\tilde{\mathbf{u}} \in \tilde{Q}$, all $\mathbf{x} \in \mathcal{X}$ s.t. $\mathcal{G}_{0,\mathbf{w},\mathbf{u}}(\mathbf{x}) \neq y$, we have

$$
\begin{aligned}
f_{\mathbf{w}+\tilde{\mathbf{u}}}(\mathbf{x})[\mathcal{G}_{0,\mathbf{w},\mathbf{u}}(\mathbf{x})] + \frac{\gamma}{4} &> f_{\mathbf{w}}(\mathbf{x})[\mathcal{G}_{0,\mathbf{w},\mathbf{u}}(\mathbf{x})] + \frac{\gamma}{8} \\
&> f_{\mathbf{w}+\check{\mathbf{u}}}(\mathbf{x}+\check{\mathbf{u}})[\mathcal{G}_{0,\mathbf{w},\mathbf{u}}(\mathbf{x})] \\
&> f_{\mathbf{w}+\check{\mathbf{u}}}(\mathbf{x}+\check{\mathbf{u}})[y] \\
&> f_{\mathbf{w}}(\mathbf{x})[y] - \frac{\gamma}{8} \\
&> f_{\mathbf{w}+\tilde{\mathbf{u}}}(\mathbf{x})[y] - \frac{\gamma}{4}.
\end{aligned}
\tag{19}
$$

Thus we have $\mathcal{L}_0(\mathcal{G}) \leq \mathcal{L}_{\frac{\gamma}{2}}(f_{\mathbf{w},\tilde{\mathbf{u}}})$. $\qquad\square$

**Proof B.2** *Given (14), (15) and (17), for all $\tilde{\mathbf{u}} \in \tilde{Q}$, if there exists $\mathbf{x} \in \mathcal{X}$ and $\check{\mathbf{u}} \in \tilde{Q}$ s.t. $f_{\mathbf{w}+\check{\mathbf{u}}}(\mathbf{x})[y] < \max_{j\neq y} f_{\mathbf{w}+\check{\mathbf{u}}}(\mathbf{x})[j] + \frac{\gamma}{2}$, we have*

$$
\begin{aligned}
f_{\mathbf{w}+\tilde{\mathbf{u}}}(\mathbf{x}+\tilde{\mathbf{u}})[j] + \frac{3\gamma}{4} &> f_{\mathbf{w}}(\mathbf{x})[j] + \frac{5\gamma}{8} \\
&> f_{\mathbf{w}+\check{\mathbf{u}}}(\mathbf{x})[j] + \frac{\gamma}{2} \\
&> f_{\mathbf{w}+\check{\mathbf{u}}}(\mathbf{x})[y] \\
&> f_{\mathbf{w}}(\mathbf{x})[y] - \frac{\gamma}{8} \\
&> f_{\mathbf{w}+\tilde{\mathbf{u}}}(\mathbf{x}+\tilde{\mathbf{u}})[y] - \frac{\gamma}{4}.
\end{aligned}
\tag{20}
$$

*Hence, if there exists $\mathbf{x} \in \mathcal{X}$ and $\check{\mathbf{u}} \in \tilde{Q}$ s.t. $f_{\mathbf{w}+\check{\mathbf{u}}}(\mathbf{x})[y] < \max_{j\neq y} f_{\mathbf{w}+\check{\mathbf{u}}}(\mathbf{x})[j] + \frac{\gamma}{2}$, we have $f_{\mathbf{w}+\tilde{\mathbf{u}}}(\mathbf{x}+\tilde{\mathbf{u}})[j] + \gamma > f_{\mathbf{w}+\tilde{\mathbf{u}}}(\mathbf{x}+\tilde{\mathbf{u}})[y]$ for all $\tilde{\mathbf{u}} \in \tilde{Q}$. Given $z \geq \frac{1}{2}$, we have $\mathcal{G}_{\gamma,\mathbf{w},\mathbf{u}}(\mathbf{x}) \neq y$. Thus, we have $\widehat{\mathcal{L}}_\gamma(\mathcal{G}) \geq \widehat{\mathcal{L}}_{\frac{\gamma}{2}}(f_{\mathbf{w},\tilde{\mathbf{u}}})$.* $\qquad\square$

**Proof B.3** *Given $q$, $\tilde{q}$, $z$ and $\mathcal{S}_{\mathbf{u}}$ in (15), let $\mathcal{S}_{\mathbf{u}}^c$ denote the complement set of $\mathcal{S}_{\mathbf{u}}$ and $\tilde{q}^c$ denote the normalized density function restricted to $\mathcal{S}_{\mathbf{u}}^c$. Then, we have*

$$
D_{\mathrm{KL}}(q\|p) = z D_{\mathrm{KL}}(\tilde{q}\|p) + (1-z) D_{\mathrm{KL}}(\tilde{q}^c\|p) - H(z),
\tag{21}
$$

*where $H(z) = -z\ln z - (1-z)\ln(1-z) \leq 1$ is the binary entropy function. Since $D_{\mathrm{KL}}$ is always positive, we get*

$$
D_{\mathrm{KL}}(\tilde{q}\|p) = \frac{1}{z}[D_{\mathrm{KL}}(q\|p) + H(z)) - (1-z)D_{\mathrm{KL}}(\tilde{q}^c\|p)] \leq 2(D_{\mathrm{KL}}(q\|p) + 1).
\tag{22}
$$

*Thus we have $2(D_{\mathrm{KL}}(\mathbf{w}+\mathbf{u}\|P) + \log\frac{6m}{\delta}) \geq D_{\mathrm{KL}}(\mathbf{w}+\tilde{\mathbf{u}}\|P) + \log\frac{2m}{\delta}$.* $\qquad\square$

## B.2 PROOF FOR THM. 3.2

First, we restate Thm. 3.2 here.

**Theorem 3.2** *Given Lem. 3.1, for any $B, n, h > 0$, let $f_{\mathbf{w}} : \mathcal{X} \to \mathcal{Y}$ be an $n$-layer feedforward network with $h$ units each layer and ReLU activation function. Choose the largest perturbation $\mathbf{u}$ under the restriction, for any $\delta, \gamma > 0$, any $\mathbf{w}$ over training dataset of size $m$, with probability at least $1 - \delta$, we have the the following bound:*

$$
\mathcal{L}_0(\mathcal{G}) \leq \widehat{\mathcal{L}}_\gamma(\mathcal{G}) + \mathcal{O}\left(\sqrt{\frac{\Phi\left(\prod_i \|\mathbf{W}_i\|_2^2, \sum_i \|\mathbf{W}_i\|_F^2\right) + \ln\frac{nm}{\delta}}{m-1}}\right),
$$

*where*

$$
\Phi\left(\prod_i \|\mathbf{W}_i\|_2^2, \sum_i \|\mathbf{W}_i\|_F^2\right) = \frac{\sum_i \left(\|\mathbf{W}_i\|_F^2 / \|\mathbf{W}_i\|_2^2\right)}{\Psi\left(\prod_i \|\mathbf{W}_i\|_2^2\right) / \left(\prod_i \|\mathbf{W}\|_2^2\right)^{\frac{1}{n}}},
$$

*and*

$$\Psi\left(\prod_i \|\mathbf{W}_i\|_2^2\right) = \left(\left(\gamma \middle/ \left(2^8 n(h\ln(8nh))^{\frac{1}{2}}\tau^{\frac{1}{2}}\prod_i \|\mathbf{W}_i\|_2^{\frac{n-1}{n}}\right)\right) + \frac{B^2}{4\tau}\right)^{\frac{1}{2}} - \frac{B}{2\tau^{\frac{1}{2}}}\right)^2.$$

*Here $\tau$ is the solution of $F_{\chi_d^2}(\tau) = \frac{\sqrt{2}}{2}$, where $F_{\chi_d^2}(\cdot)$ is the CDF for the chi-square distribution $\chi_d^2$ with $d$ degrees of freedom.*

In the following, we provide the details of the proof.

**Proof for Thm. 3.2.** *Following Neyshabur et al. (2017b), we also use two main steps to prove Thm. 3.2. Firstly, utilizing Lems. B.4 and B.5, we compute the maximum allowable perturbation of $\mathbf{u}$ required to satisfy the given condition on the margin $\gamma$. In the second step, we compute the KL term in Lem. 3.1, considering the perturbation obtained from the previous step. This computation is essential in deriving the PAC-Bayesian bound.*

*Consider a network with weights $\mathbf{W}$ that we can regularize through dividing each weight matrix $\mathbf{W}_i$ by its spectral norm $\|\mathbf{W}_i\|_2$. Let $\beta$ be defined as the geometric mean of the spectral norms of all weight matrices, i.e., $\beta = \left(\prod_{i=1}^n \|\mathbf{W}_i\|_2\right)^{1/n}$. We introduce a modified version of the weights, denoted as $\widetilde{\mathbf{W}}_i = \frac{\beta}{\|\mathbf{W}_i\|_2}\mathbf{W}_i$, which is obtained by scaling the original weights $\mathbf{W}_i$ by a factor of $\frac{\beta}{\|\mathbf{W}_i\|_2}$. As a consequence of the homogeneity property of ReLU, the behavior of the network with the modified weights $f_{\tilde{\mathbf{w}}}$, is the same as the original network $f_{\mathbf{w}}$.*

*Moreover, we find that the product of the spectral norms of the original weights, $(\prod_{i=1}^n \|\mathbf{W}_i\|_2)$, is equal to the product of the spectral norms of the modified weights, $\left(\prod_{i=1}^n \left\|\widetilde{\mathbf{W}}_i\right\|_2\right)$. Additionally, the ratio of the Frobenius norm of the original weights to their spectral norm is equal to the ratio of the modified weights, i.e., $\frac{\|\mathbf{W}_i\|_F}{\|\mathbf{W}_i\|_2} = \frac{\|\tilde{\mathbf{W}}_i\|_F}{\|\tilde{\mathbf{W}}_i\|_2}$. Consequently, the excess error mentioned in the Theorem statement remains unchanged under this weight normalization. Hence, it suffices to prove the Theorem only for the normalized weights $\tilde{\mathbf{w}}$. We assume, without loss of generality, that the spectral norm of each weight matrix is equal to $\beta$, i.e., for any layer $i$, $\|\mathbf{W}_i\|_2 = \beta$.*

*Now, we choose the prior distribution $P$ to be a Gaussian distribution with zero mean and a diagonal covariance matrix of $\sigma^2\mathbf{I}$. We introduce a random perturbation $\mathbf{u} = \begin{pmatrix} \mathbf{u}^{(w)} \\ \mathbf{u}^{(x)} \end{pmatrix} \sim \mathcal{N}(0, \sigma^2\mathbf{I})$, where the value of $\sigma$ will be determined based on $\beta$ later. More precisely, since the prior cannot depend on the learned predictor $\mathbf{w}$ or its norm, we will set $\sigma$ based on an approximation $\tilde{\beta}$. For each value of $\tilde{\beta}$ on a pre-determined grid, we will compute the PAC-Bayesian bound, establishing the generalization guarantee for all $\mathbf{w}$ for which $|\beta - \tilde{\beta}| \leq \frac{1}{n}\beta$, and ensuring that each relevant value of $\beta$ is covered by some $\tilde{\beta}$ on the grid. We will then take a union bound over all $\tilde{\beta}$ on the grid. For now, we will consider a fixed $\tilde{\beta}$ and the $\mathbf{w}$ for which $|\beta - \tilde{\beta}| \leq \frac{1}{n}\beta$, and hence $\frac{1}{e}\beta^{n-1} \leq \tilde{\beta}^{n-1} \leq e\beta^{n-1}$.*

*According to Bandeira & Boedihardjo (2021) and $\mathbf{u}^{(w)} \sim \mathcal{N}(0, \sigma^2\mathbf{I})$, we can get the following bound for $\|\mathbf{U}_i^{(w)}\|_2$:*

$$\mathbb{P}_{\mathbf{u}_i^{(w)} \sim \mathcal{N}(0, \sigma^2\mathbf{I})}\left[\left\|\mathbf{U}_i^{(w)}\right\|_2 > t\right] \leq 2he^{-t^2/2h\sigma^2}. \tag{23}$$

*By taking a union bound over the layers, we can establish that, with a probability $\geq \frac{\sqrt{2}}{2}$, the spectral norm of the perturbation $\mathbf{U}_i^{(w)}$ in each layer is bounded by $\sigma\sqrt{2h\ln((4+2\sqrt{2})nh)}$.*

*As $\mathbf{u}^{(x)} \sim \mathcal{N}(0, \sigma^2\mathbf{I})$ and takes value in $\mathbb{R}^d$, $\frac{\|\mathbf{u}^{(x)}\|_2^2}{\sigma^2}$ has a chi-square distribution $\chi_d^2$. Let $F_{\chi_d^2}(\cdot)$ be the cumulative distribution function of $\chi_d^2$, and $F_{\chi_d^2}(\tau) = \frac{\sqrt{2}}{2}$. Then, with a probability $\frac{\sqrt{2}}{2}$, $\|\mathbf{u}^{(x)}\|_2$ is upper bounded by $\sqrt{\tau}\sigma$.*

*Thus, with probability at least $\frac{1}{2}$ (i.e., $\frac{\sqrt{2}}{2} \cdot \frac{\sqrt{2}}{2}$), the above bounds can both hold. Plugging the bounds into Lems. B.4 and B.5, we have that*

$$
\begin{aligned}
\max_{\mathbf{x} \in \mathcal{X}} \|f_{\mathbf{w}+\mathbf{u}}(\mathbf{x}) - f_{\mathbf{w}}(\mathbf{x})\|_2 &\leq eB\beta^n \sum_i \frac{\left\|\mathbf{U}_i^{(w)}\right\|_2}{\beta} \\
&= eB\beta^{n-1} \sum_i \left\|\mathbf{U}_i^{(w)}\right\|_2 \\
&\leq e^2 nB\tilde{\beta}^{n-1}\sigma\sqrt{2h\ln((4+2\sqrt{2})nh)} \leq \frac{\gamma}{8},
\end{aligned}
\tag{24}
$$

*and*

$$
\begin{aligned}
\max_{\mathbf{x} \in \mathcal{X}} \|f_{\mathbf{w}+\mathbf{u}}(\mathbf{x}+\mathbf{u}) - f_{\mathbf{w}}(\mathbf{x})\|_2 &\leq e(B + \|\mathbf{u}^{(x)}\|_2)\beta^n \sum_i \frac{\left\|\mathbf{U}_i^{(w)}\right\|_2}{\beta} \\
&= e(B + \|\mathbf{u}^{(x)}\|_2)\beta^{n-1} \sum_i \left\|\mathbf{U}_i^{(w)}\right\|_2 \\
&\leq e^2 n(B + \sqrt{\tau}\sigma)\tilde{\beta}^{n-1}\sigma\sqrt{2h\ln((4+2\sqrt{2})nh)} \leq \frac{\gamma}{8}.
\end{aligned}
\tag{25}
$$

*To make (24) and (25) both hold, given $\tilde{\beta}^{n-1} \leq e\beta^{n-1}$, we can choose the largest $\sigma$ (with numerical simplification, i.e., $8e^3\sqrt{2} < 2^8$, $4 + 2\sqrt{2} < 8$) as*

$$
\sigma^2 = \Psi\left(\prod_i \|\mathbf{W}_i\|_2^2\right) = \left(\left(\gamma \Big/ \left(2^8 n(h\ln(8nh))^{\frac{1}{2}}\tau^{\frac{1}{2}} \prod_{i=1}^n \|\mathbf{W}_i\|_2^{\frac{n-1}{n}}\right) + \frac{B^2}{4\tau}\right)^{\frac{1}{2}} - \frac{B}{2\tau^{\frac{1}{2}}}\right)^2.
$$

*Hence, the perturbation $\mathbf{u}$ with the above value of $\sigma$ satisfies the assumptions of the Lem. 3.1. We now compute the KL-term in Lem. 3.1 using the selected distributions for $P$ and $\mathbf{u}$, considering the given value of $\sigma$.*

$$
\begin{aligned}
D_{\mathrm{KL}}(\mathbf{w}+\mathbf{u}\|P) &\leq \frac{\|\mathbf{w}\|_2^2}{2\sigma^2} \\
&= \frac{\sum_{i=1}^n \|\mathbf{W}_i\|_F^2}{2\sigma^2} \\
&\leq \mathcal{O}\left(\Phi\left(\prod_i \|\mathbf{W}_i\|_2^2, \sum_i \|\mathbf{W}_i\|_F^2\right)\right),
\end{aligned}
$$

*where*

$$
\Phi\left(\prod_i \|\mathbf{W}_i\|_2^2, \sum_i \|\mathbf{W}_i\|_F^2\right) = \frac{\sum_i \left(\|\mathbf{W}_i\|_F^2/\|\mathbf{W}_i\|_2^2\right)}{\Psi\left(\prod_i \|\mathbf{W}_i\|_2^2\right)/(\prod_i \|\mathbf{W}\|_2^2)^{\frac{1}{n}}}.
\tag{26}
$$

*Clearly, $\Phi$ increases with the growth of $\prod_i \|\mathbf{W}_i\|_2^2$ and $\sum_i \|\mathbf{W}_i\|_F^2$, since*

$$
\begin{aligned}
\frac{\partial \Phi}{\partial\left(\prod_i \|\mathbf{W}_i\|_2^2\right)} &> 0, \\
\frac{\partial \Phi}{\partial\left(\sum_i \|\mathbf{W}_i\|_F^2\right)} &> 0,
\end{aligned}
\tag{27}
$$

*for all $\prod_i \|\mathbf{W}_i\|_2^2 > 0$ and $\sum_i \|\mathbf{W}_i\|_F^2 > 0$.*

*Then, we can give a union bound over different choices of $\tilde{\beta}$. We only need to form the bound for $\left(\frac{\gamma}{2B}\right)^{\frac{1}{n}} \leq \beta \leq \left(\frac{\gamma\sqrt{m}}{2B}\right)^{\frac{1}{n}}$ which can be covered using a cover of size $nm^{\frac{1}{2n}}$ as discussed in Neyshabur et al. (2017b). Thus, with probability $\geq 1 - \delta$, for any $\tilde{\beta}$ and for all $\mathbf{w}$ such that $|\beta - \tilde{\beta}| \leq \frac{1}{n}\beta$, we have:*

$$
\mathcal{L}_0(\mathcal{G}) \leq \hat{\mathcal{L}}_\gamma(\mathcal{G}) + \mathcal{O}\left(\sqrt{\frac{\Phi\left(\prod_i \|\mathbf{W}_i\|_2^2, \sum_i \|\mathbf{W}_i\|_F^2\right) + \ln\frac{nm}{\delta}}{m - 1}}\right).
$$

*Hence, proved.* □

**Lemma B.4 (Neyshabur et al. (2017b))** *For any $B, n > 0$, let $f_{\mathbf{w}} : \mathcal{X} \to \mathcal{Y}$ be a n-layer feedforward network with ReLU activation function. Then for any $\mathbf{w}$, and $\mathbf{x} \in \mathcal{X}$, and any perturbation $\mathbf{u}^{(w)} = vec(\{\mathbf{U}_i^{(w)}\}_{i=1}^n)$ such that $\|\mathbf{U}_i^{(w)}\|_2 \leq \frac{1}{n}\|\mathbf{W}_i\|_2$, the change in the output of the network can be bounded as follow*

$$\|f_{\mathbf{w}+\mathbf{u}}(\mathbf{x}) - f_{\mathbf{w}}(\mathbf{x})\|_2 \leq eB \left(\prod_{i=1}^n \|\mathbf{W}_i\|_2\right) \sum_{i=1}^n \frac{\left\|\mathbf{U}_i^{(w)}\right\|_2}{\|\mathbf{W}_i\|_2}. \tag{28}$$

**Proof for Lem. B.4. (Neyshabur et al. (2017b))** *Let $\Delta_i = \|f_{\mathbf{w}+\mathbf{u}}^i(\mathbf{x}) - f_{\mathbf{w}}^i(\mathbf{x})\|_2$, we will prove using induction that for any $i \geq 0$:*

$$\Delta_i \leq \left(1 + \frac{1}{n}\right)^i \left(\prod_{j=1}^i \|\mathbf{W}_j\|_2\right) \|\mathbf{x}\|_2 \sum_{j=1}^i \frac{\left\|\mathbf{U}_j^{(w)}\right\|_2}{\|\mathbf{W}_j\|_2}. \tag{29}$$

*The above inequality together with $(1 + \frac{1}{n})^n \leq e$ proves the lemma statement. The induction base clearly holds since . For any $i \geq 1$, we have the following*

$$\begin{aligned}
\Delta_{i+1} &= \left\|\left(\mathbf{W}_{i+1} + \mathbf{U}_{i+1}^{(w)}\right)\phi\left(f_{\mathbf{w}+\mathbf{u}}^i(\mathbf{x})\right) - \mathbf{W}_{i+1}\phi\left(f_{\mathbf{w}}^i(\mathbf{x})\right)\right\|_2 \\
&= \left\|\left(\mathbf{W}_{i+1} + \mathbf{U}_{i+1}^{(w)}\right)\left(\phi\left(f_{\mathbf{w}+\mathbf{u}}^i(\mathbf{x})\right) - \phi\left(f_{\mathbf{w}}^i(\mathbf{x})\right)\right) + \mathbf{U}_{i+1}^{(w)}\phi\left(f_{\mathbf{w}}^i(\mathbf{x})\right)\right\|_2 \\
&\leq \left(\|\mathbf{W}_{i+1}\|_2 + \left\|\mathbf{U}_{i+1}^{(w)}\right\|_2\right)\left\|\phi\left(f_{\mathbf{w}+\mathbf{u}}^i(\mathbf{x})\right) - \phi\left(f_{\mathbf{w}}^i(\mathbf{x})\right)\right\|_2 + \left\|\mathbf{U}_{i+1}^{(w)}\right\|_2 \left\|\phi\left(f_{\mathbf{w}}^i(\mathbf{x})\right)\right\|_2 \\
&\leq \left(\|\mathbf{W}_{i+1}\|_2 + \left\|\mathbf{U}_{i+1}^{(w)}\right\|_2\right)\left\|f_{\mathbf{w}+\mathbf{u}}^i(\mathbf{x}) - f_{\mathbf{w}}^i(\mathbf{x})\right\|_2 + \left\|\mathbf{U}_{i+1}^{(w)}\right\|_2 \left\|f_{\mathbf{w}}^i(\mathbf{x})\right\|_2 \\
&= \Delta_i \left(\|\mathbf{W}_{i+1}\|_2 + \left\|\mathbf{U}_{i+1}^{(w)}\right\|_2\right) + \left\|\mathbf{U}_{i+1}^{(w)}\right\|_2 \left\|f_{\mathbf{w}}^i(\mathbf{x})\right\|_2,
\end{aligned}$$

*where the last inequality is by the Lipschitz property of the activation function and using $\phi(0) = 0$. The $\ell_2$ norm of outputs of layer $i$ is bounded by $\|\mathbf{x}\|_2 \Pi_{j=1}^i \|\mathbf{W}_j\|_2$ and by the lemma assumption we have $\|\mathbf{U}_{i+1}^{(w)}\|_2 \leq \frac{1}{n}\|\mathbf{W}_{i+1}\|_2$. Therefore, using the induction step, we get the following bound:*

$$\begin{aligned}
\Delta_{i+1} &\leq \Delta_i \left(1 + \frac{1}{n}\right)\|\mathbf{W}_{i+1}\|_2 + \left\|\mathbf{U}_{i+1}^{(w)}\right\|_2 \|\mathbf{x}\|_2 \prod_{j=1}^i \|\mathbf{W}_j\|_2 \\
&\leq \left(1 + \frac{1}{n}\right)^{i+1}\left(\prod_{j=1}^{i+1}\|\mathbf{W}_j\|_2\right)\|\mathbf{x}\|_2 \sum_{j=1}^i \frac{\left\|\mathbf{U}_j^{(w)}\right\|_2}{\|\mathbf{W}_j\|_2} + \frac{\left\|\mathbf{U}_{i+1}^{(w)}\right\|_2}{\|\mathbf{W}_{i+1}\|_2}\|\mathbf{x}\|_2 \prod_{j=1}^{i+1}\|\mathbf{W}_i\|_2 \quad (30) \\
&\leq \left(1 + \frac{1}{n}\right)^{i+1}\left(\prod_{j=1}^{i+1}\|\mathbf{W}_j\|_2\right)\|\mathbf{x}\|_2 \sum_{j=1}^{i+1}\frac{\|\mathbf{U}_j\|_2}{\|\mathbf{W}_j\|_2}.
\end{aligned}$$

*Hence, proved.* □

**Lemma B.5** *Given Lem. B.4 and the proof, we have*

$$\|f_{\mathbf{w}+\mathbf{u}}(\mathbf{x}+\mathbf{u}) - f_{\mathbf{w}}(\mathbf{x})\|_2 \leq e(B + \|\mathbf{u}^{(x)}\|_2)\left(\prod_{i=1}^n \|\mathbf{W}_i\|_2\right)\sum_{i=1}^n \frac{\left\|\mathbf{U}_i^{(w)}\right\|_2}{\|\mathbf{W}_i\|_2}. \tag{31}$$

### B.3 PROOF FOR THM. 3.3

First, we restate Thm. 3.3 here.

**Theorem 3.3** *Given Thm. 3.2, for any $\mathbf{x} \in \mathcal{X}$, suppose there exist $p_{\mathbf{w},\mathbf{u}}^A(\mathbf{x})$, $p_{\mathbf{w},\mathbf{u}}^B(\mathbf{x})$ such that*

$$\mathbb{E}_{\mathbf{u}} \mathbb{1}\left[\arg\max_c f_{\mathbf{w}+\mathbf{u}}(\mathbf{x}+\mathbf{u})[c] = \mathcal{G}_{0,\mathbf{w},\mathbf{u}}(\mathbf{x})\right] \geq p_{\mathbf{w},\mathbf{u}}^A(\mathbf{x}) \geq p_{\mathbf{w},\mathbf{u}}^B(\mathbf{x})$$

$$\geq \max_{j \neq \mathcal{G}_{0,\mathbf{w},\mathbf{u}}(\mathbf{x})} \mathbb{E}_{\mathbf{u}} \mathbb{1}\left[\arg\max_c f_{\mathbf{w}+\mathbf{u}}(\mathbf{x}+\mathbf{u})[c] = j\right].$$

*Then, for any $\delta, \gamma > 0$, with probability at least $1 - \delta$ we have*

$$\mathcal{L}_0(\mathcal{G}, \epsilon) \leq \widehat{\mathcal{L}}_\gamma(\mathcal{G}) + \mathcal{O}\left(\sqrt{\frac{\Phi\left(\prod_i \|\mathbf{W}_i\|_2^2, \sum_i \|\mathbf{W}_i\|_F^2\right) + \ln\frac{nm}{\delta}}{m - 1}}\right)$$

*within $\ell_2$ norm data perturbation radius $\sqrt{\epsilon_{\mathbf{x}}}$, where*

$$\epsilon_{\mathbf{x}} = \underbrace{-\ln\left(1 - \left(\sqrt{p_{\mathbf{w},\mathbf{u}}^A(\mathbf{x})} - \sqrt{p_{\mathbf{w},\mathbf{u}}^B(\mathbf{x})}\right)^2\right)}_{\textit{Model and Data Joint Dependence}} \cdot 2\,\Psi\underbrace{\left(\prod_i \|\mathbf{W}_i\|_2^2\right)}_{\textit{Model Dependence}}.$$

In the following, we provide the details of the proof. This proof is developed from Dvijotham et al. (2020), but their smoothing function is only applied on $\mathbf{x}$, our certification is with respect to the smoothing scheme on both $\mathbf{w}$ and $\mathbf{x}$.

**Proof for Thm. 3.3.** *Consider a classifier $f_{\mathbf{w}} : \mathcal{X} \to \mathcal{Y}$. The output of the classifier relies on both the input $\mathbf{x}$ and its model weights $\mathbf{w}$. We would like to verify the robustness of smoothed classifier $\mathcal{G}$. Recall that we smooth the classify on both $\mathbf{x}$ and $\mathbf{w}$ with $\mathbf{u} = \begin{pmatrix} \mathbf{u}^{(w)} \\ \mathbf{u}^{(x)} \end{pmatrix} \sim \mathcal{N}(0, \sigma^2\mathbf{I})$, to prove Thm. 3.3, our goal is to certify that*

$$\mathcal{G}_{0,\mathbf{w},\mathbf{u}}(\mathbf{x}) = \mathcal{G}_{0,\mathbf{w},\mathbf{u}}(\mathbf{x} + \varepsilon) \tag{32}$$

*for all $\varepsilon \in \{\varepsilon \in \mathbb{R}^d \mid \|\varepsilon\|_2^2 \leq \epsilon_{\mathbf{x}}\}$ and all $\mathbf{x} \in \mathcal{X}$, where $\epsilon_{\mathbf{x}}$ satisfies the condition in Thm. 3.3.*

*First, recall that we use $\mathbf{w} + \mathbf{u}$ and $\mathbf{x} + \mathbf{u}$ to denote $\mathbf{w} + \mathbf{u}^{(w)}$ and $\mathbf{x} + \mathbf{u}^{(x)}$, respectively. We let $D_g$ be g-divergence (as we have used $f(\cdot)$, we define g-divergence rather than f-divergence), $\epsilon_g(\mathbf{x}) = D_g(\nu\|\rho)$ where $\nu$ is the joint distribution of $\mathbf{w} + \mathbf{u}$ and $\mathbf{x} + \varepsilon + \mathbf{u}$ with probability density function (PDF) $\nu(\cdot, \cdot)$, $\rho$ is the joint distribution of $\mathbf{w} + \mathbf{u}$ and $\mathbf{x} + \mathbf{u}$ with PDF $\rho(\cdot, \cdot)$, $p_{\mathbf{w},\mathbf{u}}^A(\mathbf{x})$ and $p_{\mathbf{w},\mathbf{u}}^B(\mathbf{x})$ be as in Thm. 3.3. Let*

$$\mathcal{P}_{\mathbf{w},\mathbf{u}}^A(\mathbf{x}) = \mathbb{E}_{\mathbf{u}} \mathbb{1}\left[\arg\max_c f_{\mathbf{w}+\mathbf{u}}(\mathbf{x}+\mathbf{u})[c] = \mathcal{G}_{0,\mathbf{w},\mathbf{u}}(\mathbf{x})\right],$$

$$\mathcal{P}_{\mathbf{w},\mathbf{u}}^B(\mathbf{x}) = \max_{j \neq \mathcal{G}_{0,\mathbf{w},\mathbf{u}}(\mathbf{x})} \mathbb{E}_{\mathbf{u}} \mathbb{1}\left[\arg\max_c f_{\mathbf{w}+\mathbf{u}}(\mathbf{x}+\mathbf{u})[c] = j\right], \tag{33}$$

$$\mathcal{P}_{\mathbf{w},\mathbf{u}}^A(\mathbf{x}) + \mathcal{P}_{\mathbf{w},\mathbf{u}}^B(\mathbf{x}) \leq 1,$$

*and*

$$\mathcal{P}_{\mathbf{w},\mathbf{u}}^A(\mathbf{x}) \geq p_{\mathbf{w},\mathbf{u}}^A(\mathbf{x}) \geq p_{\mathbf{w},\mathbf{u}}^B(\mathbf{x}) \geq \mathcal{P}_{\mathbf{w},\mathbf{u}}^B(\mathbf{x}). \tag{34}$$

*According to Pf. B.6, we have that: the smoothed classifier $\mathcal{G}_{0,\mathbf{w},\mathbf{u}}(\mathbf{x})$ is robustly certified, i.e., (32) holds, if the optimal value of the following convex optimization problem is non-negative, i.e.,*

$$\max_{\lambda \geq 0, \kappa} \kappa - \lambda\epsilon_g(\mathbf{x}) - \mathcal{P}_{\mathbf{w},\mathbf{u}}^A(\mathbf{x})g_\lambda^*(\kappa-1) - \mathcal{P}_{\mathbf{w},\mathbf{u}}^B(\mathbf{x})g_\lambda^*(\kappa+1) - (1 - \mathcal{P}_{\mathbf{w},\mathbf{u}}^A(\mathbf{x}) - \mathcal{P}_{\mathbf{w},\mathbf{u}}^B(\mathbf{x}))g_\lambda^*(\kappa) \geq 0,$$

$$\tag{35}$$

*where $g_\lambda^*(u) = \max_{v \geq 0}(uv - g_\lambda(v))$, $g_\lambda(v) = \lambda g(v)$, the function $g(\cdot)$ is used in g-divergence.*

*Then, let $D_g$ be $D_{\mathrm{KL}}$, according to Pf. B.7, the optimization problem of (35) is non-negative if*

$$D_{\mathrm{KL}}(\nu\|\rho) \leq -\ln\left(1 - \left(\sqrt{\mathcal{P}_{\mathbf{w},\mathbf{u}}^A(\mathbf{x})} - \sqrt{\mathcal{P}_{\mathbf{w},\mathbf{u}}^B(\mathbf{x})}\right)^2\right). \tag{36}$$

*Since* $\mathbf{u} = \begin{pmatrix} \mathbf{u}^{(w)} \\ \mathbf{u}^{(x)} \end{pmatrix} \sim \mathcal{N}(0, \sigma^2\mathbf{I})$, *we have*

$$
\begin{aligned}
D_{\mathrm{KL}}(\nu\|\rho) &= D_{\mathrm{KL}}\left( \begin{pmatrix} \mathbf{w}+\mathbf{u}^{(w)} \\ \mathbf{x}+\boldsymbol{\varepsilon}+\mathbf{u}^{(x)} \end{pmatrix} \middle\| \begin{pmatrix} \mathbf{w}+\mathbf{u}^{(w)} \\ \mathbf{x}+\mathbf{u}^{(x)} \end{pmatrix} \right) \\
&= \frac{\| \left( \begin{smallmatrix} \mathbf{w} \\ \mathbf{x}+\boldsymbol{\varepsilon} \end{smallmatrix} \right) - \left( \begin{smallmatrix} \mathbf{w} \\ \mathbf{x} \end{smallmatrix} \right) \|_2^2}{2\sigma^2} \\
&= \frac{\|\boldsymbol{\varepsilon}\|_2^2}{2\sigma^2}.
\end{aligned}
\tag{37}
$$

*Given Thm. 3.2, (34), (36) and (37), we have that* $\mathcal{G}_{0,\mathbf{w},\mathbf{u}}(\mathbf{x})$ *is certified robust if*

$$
\|\boldsymbol{\varepsilon}\|_2^2 \le \epsilon_{\mathbf{x}} = \underbrace{-\ln\left(1 - \left(\sqrt{p_{\mathbf{w},\mathbf{u}}^A(\mathbf{x})} - \sqrt{p_{\mathbf{w},\mathbf{u}}^B(\mathbf{x})}\right)^2\right)}_{\textit{Data Dependence}} \cdot \underbrace{2\,\Psi\left(\prod_i \|\mathbf{W}_i\|_2^2\right)}_{\textit{Model Dependence}}.
\tag{38}
$$

*Hence, proved.* □

**Proof B.6** *Let $D_g$ be g-divergence, the function $g(\cdot)$ is used in g-divergence, $\epsilon_g(\mathbf{x}) = D_g(\nu\|\rho)$ where $\nu$ is the joint distribution of $\mathbf{w}+\mathbf{u}$ and $\mathbf{x}+\boldsymbol{\varepsilon}+\mathbf{u}$ with PDF $\nu(\boldsymbol{w},\boldsymbol{x})$, $\rho$ is the joint distribution of $\mathbf{w}+\mathbf{u}$ and $\mathbf{x}+\mathbf{u}$ with PDF $\rho(\boldsymbol{w},\boldsymbol{x})$, $p_{\mathbf{w},\mathbf{u}}^A(\mathbf{x})$ and $p_{\mathbf{w},\mathbf{u}}^B(\mathbf{x})$ be as in Thm. 3.3. Let $r(\boldsymbol{w},\boldsymbol{x}) = \frac{\nu(\boldsymbol{w},\boldsymbol{x})}{\rho(\boldsymbol{w},\boldsymbol{x})}$ be likelihood ratio, and*

$$
\phi(\boldsymbol{w},\boldsymbol{x}) = \begin{cases} +1, & \textit{if } \arg\max_c f_{\boldsymbol{w}}(\boldsymbol{x})[c] = \mathcal{G}_{0,\mathbf{w},\mathbf{u}}(\mathbf{x}) \\ -1, & \textit{else if } \arg\max_c f_{\boldsymbol{w}}(\boldsymbol{x})[c] = \max_{j\neq\mathcal{G}_{0,\mathbf{w},\mathbf{u}}(\mathbf{x})} \mathbb{E}_{\mathbf{u}}\mathbb{1}\left[\arg\max_c f_{\mathbf{w}+\mathbf{u}}(\mathbf{x}+\mathbf{u})[c] = j\right] \\ 0, & \textit{otherwise} \end{cases}
\tag{39}
$$

*we have*

$$
\begin{aligned}
\mathbb{E}_{(\boldsymbol{w},\boldsymbol{x})\sim\nu}[\phi(\boldsymbol{w},\boldsymbol{x})] &= \mathbb{E}_{(\boldsymbol{w},\boldsymbol{x})\sim\rho}[r(\boldsymbol{w},\boldsymbol{x})\phi(\boldsymbol{w},\boldsymbol{x})], \\
D_g(\nu\|\rho) &= \mathbb{E}_{(\boldsymbol{w},\boldsymbol{x})\sim\rho}[g(r(\boldsymbol{w},\boldsymbol{x}))], \\
\mathbb{E}_{(\boldsymbol{w},\boldsymbol{x})\sim\rho}[r(\boldsymbol{w},\boldsymbol{x})] &= 1.
\end{aligned}
\tag{40}
$$

*The third condition is obtained using the fact that $\nu$ is a probability measure. The optimization over $\nu$, which is equivalent to optimize over $r$, can be rewritten as*

$$
\begin{aligned}
&\min_{r\ge 0} \mathbb{E}_{(\boldsymbol{w},\boldsymbol{x})\sim\rho}[r(\boldsymbol{w},\boldsymbol{x})\phi(\boldsymbol{w},\boldsymbol{x})] \\
&s.t.\ \mathbb{E}_{(\boldsymbol{w},\boldsymbol{x})\sim\rho}[g(r(\boldsymbol{w},\boldsymbol{x}))] \le \epsilon_g(\mathbf{x}),\ \mathbb{E}_{(\boldsymbol{w},\boldsymbol{x})\sim\rho}[r(\boldsymbol{w},\boldsymbol{x})] = 1.
\end{aligned}
\tag{41}
$$

*We solve the optimization using Lagrangian duality as follows. We first dualize the constraints on $r$ to obtain*

$$
\begin{aligned}
&\min_{r\ge 0} \mathbb{E}_{(\boldsymbol{w},\boldsymbol{x})\sim\rho}[r(\boldsymbol{w},\boldsymbol{x})\phi(\boldsymbol{w},\boldsymbol{x})] + \lambda\left(\mathbb{E}_{(\boldsymbol{w},\boldsymbol{x})\sim\rho}[g(r(\boldsymbol{w},\boldsymbol{x}))] - \epsilon_g(\mathbf{x})\right) + \kappa\left(1 - \mathbb{E}_{(\boldsymbol{w},\boldsymbol{x})\sim\rho}[r(\boldsymbol{w},\boldsymbol{x})]\right) \\
&= \min_{r\ge 0} \mathbb{E}_{(\boldsymbol{w},\boldsymbol{x})\sim\rho}[r(\boldsymbol{w},\boldsymbol{x})\phi(\boldsymbol{w},\boldsymbol{x}) + \lambda g(r(\boldsymbol{w},\boldsymbol{x})) - \kappa r(\boldsymbol{w},\boldsymbol{x})] + \kappa - \lambda\epsilon_g(\mathbf{x}).
\end{aligned}
\tag{42}
$$

*As the $\boldsymbol{w}$ components of $\nu$ and $\rho$ are identical, but $\boldsymbol{x}$ components of $\nu$ and $\rho$ can be different, let $r(\boldsymbol{x}) = \mathbb{E}_{\boldsymbol{w}\sim\rho_{\boldsymbol{w}}}[r(\boldsymbol{w},\boldsymbol{x})]$ where $\rho_{\boldsymbol{w}}$ is the marginal distribution of $\boldsymbol{w}$ for $\rho$, the above optimization problem can be rewritten as*

$$
\begin{aligned}
&\kappa - \lambda\epsilon_g(\mathbf{x}) - \mathbb{E}_{\boldsymbol{x}\sim\rho_{\boldsymbol{x}}}\left[\max_{r\ge 0} \kappa r(\boldsymbol{x}) - r(\boldsymbol{x})\mathbb{E}_{\boldsymbol{w}\sim\rho_{\boldsymbol{w}}}\phi(\boldsymbol{w},\boldsymbol{x}) - \lambda g(r(\boldsymbol{x}))\right] \\
&= \kappa - \lambda\epsilon_g(\mathbf{x}) - \mathbb{E}_{\boldsymbol{x}\sim\rho_{\boldsymbol{x}}}\left[\max_{r\ge 0} r(\boldsymbol{x})(\kappa - \mathbb{E}_{\boldsymbol{w}\sim\rho_{\boldsymbol{w}}}\phi(\boldsymbol{w},\boldsymbol{x})) - \lambda g(r(\boldsymbol{x}))\right] \\
&= \kappa - \lambda\epsilon_g(\mathbf{x}) - \mathbb{E}_{\boldsymbol{x}\sim\rho_{\boldsymbol{x}}}\left[\max_{r\ge 0} r(\boldsymbol{x})(\kappa - \mathbb{E}_{\boldsymbol{w}\sim\rho_{\boldsymbol{w}}}\phi(\boldsymbol{w},\boldsymbol{x})) - g_\lambda(r(\boldsymbol{x}))\right] \\
&= \kappa - \lambda\epsilon_g(\mathbf{x}) - \mathbb{E}_{\boldsymbol{x}\sim\rho_{\boldsymbol{x}}}\left[g_\lambda^*(\kappa - \mathbb{E}_{\boldsymbol{w}\sim\rho_{\boldsymbol{w}}}\phi(\boldsymbol{w},\boldsymbol{x}))\right],
\end{aligned}
\tag{43}
$$

where $g_\lambda^*(u) = \max_{v \geq 0}(uv - g_\lambda(v)), g_\lambda(v) = \lambda g(v)$. Since strong duality, we can maximize the final term in (43) with respect to $\lambda \geq 0$, $\kappa$, to achieve the optimal value in (41). If the optimal value, i.e.,

$$\max_{\lambda \geq 0, \kappa} \kappa - \lambda \epsilon_g(\mathbf{x}) - \mathbb{E}_{\boldsymbol{x} \sim \rho_{\boldsymbol{x}}} \left[ g_\lambda^*(\kappa - \mathbb{E}_{\boldsymbol{w} \sim \rho_{\boldsymbol{w}}} \phi(\boldsymbol{w}; \boldsymbol{x})) \right] \tag{44}$$

is non-negative, then (32) holds. As we have

$$\max_{\lambda \geq 0, \kappa} \kappa - \lambda \epsilon_g(\mathbf{x}) - \mathbb{E}_{\boldsymbol{x} \sim \rho_{\boldsymbol{x}}} \left[ g_\lambda^*(\kappa - \mathbb{E}_{\boldsymbol{w} \sim \rho_{\boldsymbol{w}}} \phi(\boldsymbol{w}; \boldsymbol{x})) \right]$$
$$\geq \max_{\lambda \geq 0, \kappa} \kappa - \lambda \epsilon_g(\mathbf{x}) - \mathbb{E}_{(\boldsymbol{w}, \boldsymbol{x}) \sim \rho} \left[ g_\lambda^*(\kappa - \phi(\boldsymbol{w}; \boldsymbol{x})) \right], \tag{45}$$

given (33), (44), (45), we get that (32) holds if

$$\max_{\lambda \geq 0, \kappa} \kappa - \lambda \epsilon_g(\mathbf{x}) - \mathcal{P}_{\mathbf{w},\mathbf{u}}^A(\mathbf{x}) g_\lambda^*(\kappa - 1) - \mathcal{P}_{\mathbf{w},\mathbf{u}}^B(\mathbf{x}) g_\lambda^*(\kappa + 1) - (1 - \mathcal{P}_{\mathbf{w},\mathbf{u}}^A(\mathbf{x}) - \mathcal{P}_{\mathbf{w},\mathbf{u}}^B(\mathbf{x})) g_\lambda^*(\kappa) \geq 0. \tag{46}$$

$\square$

**Proof B.7** *We use the KL divergence function $g(u) = u \ln(u)$ for $D_g$, which is a convex function with $g(1) = 0$. Thus, we have*

$$g_\lambda^*(u) = \max_{v \geq 0}(uv - \lambda g(v)) = \max_{v \geq 0}(uv - \lambda v \log(v)). \tag{47}$$

*Use the derivative with respect to $v$ to 0 to solve the above optimization problem, i.e.,*

$$\frac{\partial(uv - \lambda v \ln(v))}{\partial v} = 0, \tag{48}$$

*we have $v = \ln \frac{u - \lambda}{\lambda}$, $\lambda > 0$. Thus we get*

$$g_\lambda^*(u) = \lambda \exp\left(\frac{u}{\lambda} - 1\right).$$

*Suppose there exists a bound $\epsilon_{\mathrm{KL}}(\mathbf{x})$ on the KL divergence, i.e., $D_{\mathrm{KL}}(\nu \| \rho) \leq \epsilon_{\mathrm{KL}}(\mathbf{x})$, then the optimization problem in (35) can be rewritten as*

$$\max_{\lambda > 0, \kappa} \left( \kappa - \lambda \epsilon_{\mathrm{KL}}(\mathbf{x}) - \mathcal{P}_{\mathbf{w},\mathbf{u}}^A(\mathbf{x}) \lambda \exp\left(\frac{\kappa - 1}{\lambda} - 1\right) - \mathcal{P}_{\mathbf{w},\mathbf{u}}^B(\mathbf{x}) \lambda \exp\left(\frac{\kappa + 1}{\lambda} - 1\right) \right.$$
$$\left. - (1 - \mathcal{P}_{\mathbf{w},\mathbf{u}}^A(\mathbf{x}) - \mathcal{P}_{\mathbf{w},\mathbf{u}}^B(\mathbf{x})) \lambda \exp\left(\frac{\kappa}{\lambda} - 1\right) \right) \geq 0. \tag{49}$$

*Let $\xi = \kappa/\lambda, \zeta = \frac{1}{\lambda}$ (with $\zeta > 0$), (49) can be rewritten as:*

$$\max_{\zeta > 0, \xi} \left( \frac{1}{\zeta} \Big( \xi - \epsilon_{\mathrm{KL}}(\mathbf{x}) - \mathcal{P}_{\mathbf{w},\mathbf{u}}^A(\mathbf{x}) \exp(\xi - \zeta - 1) - \mathcal{P}_{\mathbf{w},\mathbf{u}}^B(\mathbf{x}) \exp(\xi + \zeta - 1) \right.$$
$$\left. - \left(1 - \mathcal{P}_{\mathbf{w},\mathbf{u}}^A(\mathbf{x}) - \mathcal{P}_{\mathbf{w},\mathbf{u}}^B(\mathbf{x})\right) \exp(\xi - 1) \Big) \right) \geq 0. \tag{50}$$

*Since $\zeta > 0$, let both left-hand side and right-hand side time $\zeta$ and the above optimization problem can be rewritten as:*

$$\max_{\zeta > 0, \xi} \left( \xi - \epsilon_{\mathrm{KL}}(\mathbf{x}) - \mathcal{P}_{\mathbf{w},\mathbf{u}}^A(\mathbf{x}) \exp(\xi - \zeta - 1) - \mathcal{P}_{\mathbf{w},\mathbf{u}}^B(\mathbf{x}) \exp(\xi + \zeta - 1) \right.$$
$$\left. - \left(1 - \mathcal{P}_{\mathbf{w},\mathbf{u}}^A(\mathbf{x}) - \mathcal{P}_{\mathbf{w},\mathbf{u}}^B(\mathbf{x})\right) \exp(\xi - 1) \right) \geq 0. \tag{51}$$

*Setting the derivative of the left-hand side with respect to $\zeta$ to 0 and solving for $\zeta$, we obtain*

$$\mathcal{P}_{\mathbf{w},\mathbf{u}}^A(\mathbf{x}) \exp(\xi - \zeta - 1) - \mathcal{P}_{\mathbf{w},\mathbf{u}}^B(\mathbf{x}) \exp(\xi + \zeta - 1) = 0,$$
$$\zeta = \ln\left(\sqrt{\frac{\mathcal{P}_{\mathbf{w},\mathbf{u}}^A(\mathbf{x})}{\mathcal{P}_{\mathbf{w},\mathbf{u}}^B(\mathbf{x})}}\right). \tag{52}$$

*Given (51) and (52), we have*

$$\max_{\xi} \left( \xi - \epsilon_{\mathrm{KL}}(\mathbf{x}) - \left( 1 - \left( \sqrt{\mathcal{P}_{\mathbf{w},\mathbf{u}}^A(\mathbf{x})} - \sqrt{\mathcal{P}_{\mathbf{w},\mathbf{u}}^B(\mathbf{x})} \right)^2 \right) \exp(\xi - 1) \right) \geq 0. \tag{53}$$

*Setting the derivative with respect to $\xi$ to 0 and solving for $\xi$, we obtain*

$$1 - \left( 1 - \left( \sqrt{\mathcal{P}_{\mathbf{w},\mathbf{u}}^A(\mathbf{x})} - \sqrt{\mathcal{P}_{\mathbf{w},\mathbf{u}}^B(\mathbf{x})} \right)^2 \right) \exp(\xi - 1) = 0,$$

$$\xi = 1 - \ln \left( 1 - \left( \sqrt{\mathcal{P}_{\mathbf{w},\mathbf{u}}^A(\mathbf{x})} - \sqrt{\mathcal{P}_{\mathbf{w},\mathbf{u}}^B(\mathbf{x})} \right)^2 \right). \tag{54}$$

*Given (53) and (54), we have*

$$- \ln \left( 1 - \left( \sqrt{\mathcal{P}_{\mathbf{w},\mathbf{u}}^A(\mathbf{x})} - \sqrt{\mathcal{P}_{\mathbf{w},\mathbf{u}}^B(\mathbf{x})} \right)^2 \right) - \epsilon_{\mathrm{KL}}(\mathbf{x}) \geq 0. \tag{55}$$

*Then, we have*

$$D_{\mathrm{KL}}(\nu \| \rho) \leq - \ln \left( 1 - \left( \sqrt{\mathcal{P}_{\mathbf{w},\mathbf{u}}^A(\mathbf{x})} - \sqrt{\mathcal{P}_{\mathbf{w},\mathbf{u}}^B(\mathbf{x})} \right)^2 \right). \tag{56}$$

*Hence, proved.* $\qquad\qquad\square$

### B.4 PROP. 5.2

First, we state Prop. 5.2 here.

**Proposition 5.2** *Let $f_{\mathbf{w}} : \mathcal{X} \to \mathcal{Y}$ be any deterministic classifier, smoothing noise $\mathbf{u} = \begin{pmatrix} \mathbf{u}^{(w)} \\ \mathbf{u}^{(x)} \end{pmatrix} \sim \mathcal{N}(0, \sigma^2 \mathbf{I})$. For any $\mathbf{x} \in \mathcal{X}$, suppose there exists $p_{\mathbf{w},\mathbf{u}}^A(\mathbf{x})$, $p_{\mathbf{w},\mathbf{u}}^B(\mathbf{x})$ such that*

$$\mathbb{E}_{\mathbf{u}} \mathbb{1} \left[ \arg\max_c f_{\mathbf{w}+\mathbf{u}}(\mathbf{x}+\mathbf{u})[c] = \mathcal{G}_{0,\mathbf{w},\mathbf{u}}(\mathbf{x}) \right] \geq p_{\mathbf{w},\mathbf{u}}^A(\mathbf{x}) \geq p_{\mathbf{w},\mathbf{u}}^B(\mathbf{x})$$

$$\geq \max_{j \neq \mathcal{G}_{0,\mathbf{w},\mathbf{u}}(\mathbf{x})} \mathbb{E}_{\mathbf{u}} \mathbb{1} \left[ \arg\max_c f_{\mathbf{w}+\mathbf{u}}(\mathbf{x}+\mathbf{u})[c] = j \right].$$

*Then, we have $\mathcal{G}_{0,\mathbf{w},\mathbf{u}}(\mathbf{x}+\varepsilon) = \mathcal{G}_{0,\mathbf{w},\mathbf{u}}(\mathbf{x})$ for all $\|\varepsilon\|_2 \leq R$, where*

$$R^2 = -2\sigma^2 \ln \left( 1 - \left( \sqrt{p_{\mathbf{w},\mathbf{u}}^A(\mathbf{x})} - \sqrt{p_{\mathbf{w},\mathbf{u}}^B(\mathbf{x})} \right)^2 \right).$$

In the following, we provide the details of the proof.

**Proof for Prop. 5.2.** *Consider a classifier $f_{\mathbf{w}} : \mathcal{X} \to \mathcal{Y}$. The output of the classifier relies on both the input $\mathbf{x}$ and its model weights $\mathbf{w}$. We would like to verify the robustness of smoothed classifier $\mathcal{G}$. Recall that we smooth the classify on both $\mathbf{x}$ and $\mathbf{w}$ with $\mathbf{u} = \begin{pmatrix} \mathbf{u}^{(w)} \\ \mathbf{u}^{(x)} \end{pmatrix} \sim \mathcal{N}(0, \sigma^2 \mathbf{I})$, to prove Prop. 5.2, our goal is to certify that*

$$\mathcal{G}_{0,\mathbf{w},\mathbf{u}}(\mathbf{x}) = \mathcal{G}_{0,\mathbf{w},\mathbf{u}}(\mathbf{x}+\varepsilon) \tag{57}$$

*for all $\varepsilon \in \{\varepsilon \in \mathbb{R}^d \mid \|\varepsilon\|_2 \leq R\}$, where*

$$R^2 = -2\sigma^2 \ln \left( 1 - \left( \sqrt{p_{\mathbf{w},\mathbf{u}}^A(\mathbf{x})} - \sqrt{p_{\mathbf{w},\mathbf{u}}^B(\mathbf{x})} \right)^2 \right).$$

*First, recall that we use $\mathbf{w}+\mathbf{u}$ and $\mathbf{x}+\mathbf{u}$ to denote $\mathbf{w}+\mathbf{u}^{(w)}$ and $\mathbf{x}+\mathbf{u}^{(x)}$, respectively. We let $D_g$ be g-divergence (as we have used $f(\cdot)$, we define g-divergence rather than f-divergence),*

$\epsilon_g(\mathbf{x}) = D_g(\nu\|\rho)$ *where $\nu$ is the joint distribution of $\mathbf{w}+\mathbf{u}$ and $\mathbf{x}+\boldsymbol{\varepsilon}+\mathbf{u}$ with PDF $\nu(\cdot,\cdot)$, $\rho$ is the joint distribution of $\mathbf{w}+\mathbf{u}$ and $\mathbf{x}+\mathbf{u}$ with PDF $\rho(\cdot,\cdot)$, $p_{\mathbf{w},\mathbf{u}}^A(\mathbf{x})$ and $p_{\mathbf{w},\mathbf{u}}^B(\mathbf{x})$ be as in Prop. 5.2. Let*

$$\mathcal{P}_{\mathbf{w},\mathbf{u}}^A(\mathbf{x}) = \mathbb{E}_{\mathbf{u}}\mathbb{1}\Big[\arg\max_c f_{\mathbf{w}+\mathbf{u}}(\mathbf{x}+\mathbf{u})[c] = \mathcal{G}_{0,\mathbf{w},\mathbf{u}}(\mathbf{x})\Big],$$

$$\mathcal{P}_{\mathbf{w},\mathbf{u}}^B(\mathbf{x}) = \max_{j \neq \mathcal{G}_{0,\mathbf{w},\mathbf{u}}(\mathbf{x})}\mathbb{E}_{\mathbf{u}}\mathbb{1}\Big[\arg\max_c f_{\mathbf{w}+\mathbf{u}}(\mathbf{x}+\mathbf{u})[c] = j\Big], \tag{58}$$

$$\mathcal{P}_{\mathbf{w},\mathbf{u}}^A(\mathbf{x}) + \mathcal{P}_{\mathbf{w},\mathbf{u}}^B(\mathbf{x}) \leq 1,$$

*and*

$$\mathcal{P}_{\mathbf{w},\mathbf{u}}^A(\mathbf{x}) \geq p_{\mathbf{w},\mathbf{u}}^A(\mathbf{x}) \geq p_{\mathbf{w},\mathbf{u}}^B(\mathbf{x}) \geq \mathcal{P}_{\mathbf{w},\mathbf{u}}^B(\mathbf{x}). \tag{59}$$

*According to Pf. B.6, we have that: the smoothed classifier $\mathcal{G}_{0,\mathbf{w},\mathbf{u}}(\mathbf{x})$ is robustly certified, i.e., (57) holds, if the optimal value of the following convex optimization problem is non-negative, i.e.,*

$$\max_{\lambda \geq 0, \kappa} \kappa - \lambda\epsilon_g(\mathbf{x}) - \mathcal{P}_{\mathbf{w},\mathbf{u}}^A(\mathbf{x})g_\lambda^*(\kappa-1) - \mathcal{P}_{\mathbf{w},\mathbf{u}}^B(\mathbf{x})g_\lambda^*(\kappa+1) - (1-\mathcal{P}_{\mathbf{w},\mathbf{u}}^A(\mathbf{x})-\mathcal{P}_{\mathbf{w},\mathbf{u}}^B(\mathbf{x}))g_\lambda^*(\kappa) \geq 0, \tag{60}$$

*where $g_\lambda^*(u) = \max_{v \geq 0}(uv - g_\lambda(v))$, $g_\lambda(v) = \lambda g(v)$, the function $g(\cdot)$ is used in g-divergence.*

*Then, let $D_g$ be $D_{\mathrm{KL}}$, according to Pf. B.7, the optimization problem of (60) is non-negative if*

$$D_{\mathrm{KL}}(\nu\|\rho) \leq -\ln\left(1 - \left(\sqrt{\mathcal{P}_{\mathbf{w},\mathbf{u}}^A(\mathbf{x})} - \sqrt{\mathcal{P}_{\mathbf{w},\mathbf{u}}^B(\mathbf{x})}\right)^2\right). \tag{61}$$

*Since $\mathbf{u} = \begin{pmatrix}\mathbf{u}^{(w)}\\\mathbf{u}^{(x)}\end{pmatrix} \sim \mathcal{N}(0,\sigma^2\mathbf{I})$, we have*

$$\begin{aligned}
D_{\mathrm{KL}}(\nu\|\rho) &= D_{\mathrm{KL}}\left(\begin{pmatrix}\mathbf{w}+\mathbf{u}^{(w)}\\\mathbf{x}+\boldsymbol{\varepsilon}+\mathbf{u}^{(x)}\end{pmatrix}\,\Big\|\,\begin{pmatrix}\mathbf{w}+\mathbf{u}^{(w)}\\\mathbf{x}+\mathbf{u}^{(x)}\end{pmatrix}\right)\\
&= \frac{\|\begin{pmatrix}\mathbf{w}\\\mathbf{x}+\boldsymbol{\varepsilon}\end{pmatrix} - \begin{pmatrix}\mathbf{w}\\\mathbf{x}\end{pmatrix}\|_2^2}{2\sigma^2}\\
&= \frac{\|\boldsymbol{\varepsilon}\|_2^2}{2\sigma^2}.
\end{aligned} \tag{62}$$

*Given (59), (61) and (62), we have that $\mathcal{G}_{0,\mathbf{w},\mathbf{u}}(\mathbf{x})$ is certified robust if*

$$\|\boldsymbol{\varepsilon}\|_2 \leq R = \sqrt{-2\sigma^2\ln\left(1 - \left(\sqrt{p_{\mathbf{w},\mathbf{u}}^A(\mathbf{x})} - \sqrt{p_{\mathbf{w},\mathbf{u}}^B(\mathbf{x})}\right)^2\right)}. \tag{63}$$

*Hence, proved.* □

## C  EVALUATION DETAILS

In this work, our primary evaluation algorithm closely adheres to Cohen et al. (2019), while there exist two minor differences. The first difference lies in the function SAMPLEUNDERNOISE($f_{\mathbf{w}}, \mathbf{x}, num, \sigma$) of Cohen et al. (2019):

1. Draw $num$ samples of noise, $\mathbf{u}_1, ..., \mathbf{u}_{num} \sim \mathcal{N}(0,\sigma^2\mathbf{I})$.
2. Run the noisy images and noisy weights through the base classifier $f_{\mathbf{w}}$ to obtain the predictions $f_{\mathbf{w}+\mathbf{u}_1}(\mathbf{x}+\mathbf{u}_1), ..., f_{\mathbf{w}+\mathbf{u}_{num}}(\mathbf{x}+\mathbf{u}_{num})$.
3. Return the counts for each class.

Here, we smooth both the inputs and weights with $\mathbf{x}+\mathbf{u}_i$ and $\mathbf{w}+\mathbf{u}_i$, while Cohen et al. (2019) only smooths inputs.

The second difference lies in the certified robust radius, our randomized smoothing evaluation framework adopts

$$R = \sqrt{-2\sigma^2\ln\left(1 - \left(\sqrt{p_{\mathbf{w},\mathbf{u}}^A(\mathbf{x})} - \sqrt{p_{\mathbf{w},\mathbf{u}}^B(\mathbf{x})}\right)^2\right)}, \tag{64}$$

while Cohen et al. (2019) uses the inverse cumulative distribution function of Gaussian.

