# OpenReview forum: "Slightly Harmonizing Certified Robust Radius and Accuracy"
_ICLR.cc/2024/Conference — ICLR 2024 Conference Withdrawn Submission_

### Official Review · Reviewer_g8UQ · 2023-10-28

**Soundness:** 2 fair
**Presentation:** 2 fair
**Contribution:** 2 fair
**Rating:** 3
**Confidence:** 2

**Summary:**

This paper contributes theory for randomized smoothing – a method that transforms a base classifier into a certifiably robust smoothed classifier. In particular, the paper applies the PAC-Bayes framework to obtain generalization bounds for smoothed classifiers assuming the base classifier is a multi-layer perceptron with ReLU activations. Bounds are provided for regular margin loss and a robust margin loss that rewards correct predictions within the certified radius of each point. Both the bounds and certified radii are found to depend on the spectral norm of the model weights, which prompts the authors to experiment with spectral regularization. It is shown to achieve superior certified accuracy compared to non-regularized model weights in experiments.

**Strengths:**

1. This paper appears to be the first to study generalization bounds for randomized smoothing. It uses similar ideas/techniques as Farnia et al. (2019), who studied generalization bounds for adversarial training and also considered spectral regularization. Despite similarities, I think it is useful to have these results worked out for randomized smoothing.

1. The paper develops practical insights from the theory and evaluates them empirically, which adds to the impact.

**Weaknesses:**

1. The paper studies a variant of randomized smoothing that I have not seen elsewhere, which smooths the model weights in addition to the input. As a result, the theory does not apply to the large body of existing work on randomized smoothing. While proposing variations to existing methods may be beneficial, it’s important to include justification. (I could not find justification for smoothing the model weights in the paper.)

2. The experiments in Section 5.2 are missing an important baseline: randomized smoothing following Cohen et al. (2019). Currently results are only presented for the variant of randomized smoothing proposed in the paper that smooths the model weights and input. It's therefore not possible to assess whether the proposed variant improves on the baseline or not.

3. As someone without a strong background in PAC-Bayes theory, I found the paper challenging to follow. One sticking point for me is the fact that the smoothed classifier in eq (4) is not random, since the expectation is taken w.r.t $\mathbf{u}$. This lead me to wonder how PAC-Bayes is applied, given it is presented as a method for analyzing _randomized_ predictors (see p. 3). I think further explanation would be helpful in the body, even if it is ultimately covered in the proofs. Another point where further clarification would be helpful is in the interpretation of the prior and posterior on the model weights. My understanding is that the prior is the distribution used to initialize the model weights, and the posterior is the distribution of the model weights after training, however the posterior and prior do not seem to be connected via Bayes rule.

4. Some steps in the derivation of the regularizer in Section 4 are unclear. It appears the original objective is to minimize $\prod\_{i} \lVert \mathbf{W}\_{i} \rVert\_{2}^{2}$ so as to maximize the certified radius. However the original objective is replaced by a lower bound $\lVert \mathbf{W}\_{n} \cdots \mathbf{W}\_{1} \rVert\_{2}^{2}$, which is then replaced by an upper bound $\lVert \mathbf{W} \mathbf{W}^\top \rVert\_\infty$ in eqn (9). I am confused by the mixing of lower and upper bounds – normally one would minimize an upper bound on the original objective. Another step that is confusing is eqn (10). I don’t immediately see why it is true, and there is no proof provided.

Minor:
1. Eqn (4): $\mathcal{G}$ is indexed by $\mathbf{u}$, however $\mathbf{u}$ is integrated out.
1. Eqn (4): overloading $\mathbf{u}$ is confusing.
1. Section 3.1: $\Omega_r$ and $\Omega_\mathrm{ge}$ are not defined anywhere.
1. Theorem 3.3: is it correct to say that the “generalization bound holds for any data perturbation within the radius”? For this to be true, the loss must be zero for the data point in question. And as I understand it, the generalization bound is a statement about the expected loss for a randomly drawn data point.
1. Section 4: $\cos$ does not typically have two vector arguments. For clarity, should define as cosine similarity.
1. Section 4: using $\mathbf{\eta}$ where $\mathbf{u}$ was used previously.

**Questions:**

1. What is the justification for smoothing the weights in addition to the input?
1. How is PAC-Bayes applied, given the smoothed classifier is deterministic? Could you provide a proof sketch?
1. How should the posterior and prior distribution on the model weights be interpreted? Are they connected to the training algorithm?
1. Can the regularized in Section 4 be derived in a principled way, or is it a heuristic that is found to work empirically?
1. How does the certified accuracy of randomized smoothing (with weight and input smoothing) compare with vanilla randomized smoothing (input smoothing only)?

---

### Official Review · Reviewer_RAXB · 2023-10-29

**Soundness:** 3 good
**Presentation:** 3 good
**Contribution:** 3 good
**Rating:** 8
**Confidence:** 3

**Summary:**

The authors propose to apply randomized smoothing to both the inputs and the model weights simultaneously. They prove new PAC-Bayesian generalization bounds that hold over a certified robust radius within the input space. They propose a new loss term for spectral regularization of the model's weight matrices based on the insights of their generalization bounds. They show through benchmark experiments that their proposed regularization technique works, and that correspondingly it indeed increases the robust radius.

**Strengths:**

1. The author do a good job at distinguishing the contributions of their generalization bounds for smoothed classifiers from those of conventional models.
2. The theoretical results appear novel and significant.
3. The paper is well-written and easy to follow.

**Weaknesses:**

1. Since you don't prove Lemma 2.1, please provide a reference for it.
2. It is more common to use the terminology "isotropic Guassian noise" rather than "spherical Gaussian noise", so you may want to change that.
3. I suggest you replace the intersection symbol $\cap$ in Lemma 3.1 with the word "and," since you are not writing those two conditions as sets, but rather as mathematical predicates.
4. The authors make many connections to related works in the literature throughout the paper, yet, in the experiments, they do not compare their method to such related works. In my opinion, there could be more done in the experiments to further strengthen the contributions.

**Questions:**

1. It seems like you choose to use the same variance for smoothing the inputs as you do the model weights. Why do you make this restriction?

---

### Official Review · Reviewer_BZwW · 2023-10-31

**Soundness:** 2 fair
**Presentation:** 2 fair
**Contribution:** 3 good
**Rating:** 3
**Confidence:** 4

**Summary:**

This paper develops a generalization error bound for certified robustness, where the generalization bound holds under any
$\ell_2$ data perturbation within the certified robust radius. Based on their theoretical results, a spectral regularization-based method is proposed to boost the performance. Empirical studies are conducted to show the effectiveness of the proposed approach.

**Strengths:**

This paper starts a meaningful study to derive the generalization bound for certified robustness, which might bring new insights to the current ML community.

**Weaknesses:**

1. One primary concern is why we need to smooth both model weights and inputs. Although, by doing this, one can also obtain a certified region, it is not apparent that the new bound is tighter than the original one [Cohen et al., 2019]. Also, it is noteworthy that perturbing both weights and inputs would become more challenging to optimize. If it is not tight, it is meaningless. The authors should give more explanations about this and prove the tightness from the theoretical perspective.
2. With respect to the generalization bound, we should hold a strong condition $\mathbb{P}_{u}(...) \ge 1/2$. Is it achievable? In the training phase, we merely adopt the smoothed classifier to optimize. What is the relationship between $\sigma$ and this condition?
3. For the bound Thm.3.3, how to determine $\tau$ for Eq.(8)? It is recommended to validate the theoretical results by some empirical experiments.
4. Proof B.3 needs more details for easy understanding.
5. Does model weights and inputs adopt the same $\sigma$? What will happen if using a different one?
6. Finally, the empirical studies are also not unconvincing. 1) At first, I noticed that the authors leverage a different $\sigma$ value for normal and the proposed methods. Why? 2) Some sota randomized smoothing methods are overlooked [1,2,3]. The authors should conduct some ablation studies of $\sigma$ (such as $0.25, 0.5, 1.0$) and add more competitive methods. 3) Last but not least, the authors must consider some $\ell_2$-adversarial attacks to evaluate the effectiveness of the proposed method.

Ref:
- [1] MACER: attack-free and scalable robust training via maximizing certified radius.
- [2] Consistency regularization for certified robustness of smoothed classifiers.
- [3] Provably robust deep learning via adversarially trained smoothed classifiers.

**Questions:**

Please address all my concerns in the Weaknesses part.

---

### Official Review · Reviewer_fngE · 2023-10-31

**Soundness:** 3 good
**Presentation:** 2 fair
**Contribution:** 2 fair
**Rating:** 5
**Confidence:** 3

**Summary:**

The paper studies robustness and generalization guarantees for randomized-smoothing-based classifiers where the authors use weight-smoothing in addition to input-smoothing. Using the PAC-Bayes framework, the authors relate the accuracy and robustness of the proposed classifiers to the spectral norm of the neural network model. As a smaller spectral norm for the neural network leads to better guarantees, the authors suggest using a spectral norm loss to the optimization objective. In order to train neural networks with smaller spectral norms, the authors divide the spectral norm objective into a scaling and a cosine correlation term. As the weight-decay already helps to reduce the scaling term, the paper focuses on the cosine similarity term and proposes a regularizer to reduce the cosine correlation term. Finally, the authors provide some experiments to verify the empirical success of the proposed method.

**Strengths:**

The theoretical results presented in the paper are novel and provide a new relation between robustness and generalization.
The cosine correlation loss presented in the paper is shown to be empirically effective in inducing orthogonality for the weight matrix.

**Weaknesses:**

- While the initial part of the paper mainly focuses on the Pac-Bayesian bounds for generalization, the empirical section doesn't provide any validation for the non-vacuousness of the given bounds. It is hard to follow the flow of the paper as presented now. It would be great if the authors could tie in the first theoretical part better with the later parts. It is well-known that the spectral norm bounds give the effective Lipschitz constant for a given ReLU neural network model. As RS effectively measures the local Lipschitz constant of a model to give guarantees, having a known smaller global Lipschitz constant should lead to better bounds. Thus, the theoretical results do not seem to be necessary to justify the motivation for lowering the spectral norm of the network.
- The empirical results are only given for small neural network models. It is unclear if the proposed method scales well to larger neural network models. Scalability and fast inference are some of the main advantages of randomized smoothing. It is not clear if those still hold for this proposed version of the RS.

**Questions:**

1) Equation 4 is not clear. The condition for the value assignment uses $y$. What is $y$ here?

2) While it seems Theorem 3.2 and Theorem 3.3 are given to serve as a motivation for looking into reducing the spectral norm of the NN, the tightness of these bounds is not clear. It would be great to see some empirical results showing these bounds are non-vacuous in practice, and optimizing them is a good proxy for improving generalization.

3) In the empirical evaluation, it is unclear why the normal models have $\sigma=1.2$ while the spectral regularized models have $\sigma=1.3$.

4) The extra cost incurred during inference for using weight smoothing is not discussed in the paper. It would be great to see some discussion about the cost of doing both input and weight smoothing at inference time.

Notation:
- Equation 6 needs to have $\epsilon_x$ as the argument to the loss instead of $\epsilon$ as $\epsilon_x$ is used as the bound in the set builder.